# Efficient Training of Neural Stochastic Differential Equations by Matching Finite Dimensional Distributions

**Jianxin Zhang** [*]
EECS, University of Michigan
Ann Arbor, MI 48109
jianxinz@umich.edu

**Josh Viktorov, Doosan Jung, Emily Pitler**
Cisco Systems
New York, NY    Lakewood, CO    San Jose, CA
{joviktor, doojung, epitler}@cisco.com

## Abstract

Neural stochastic differential equations (Neural SDEs) have emerged as powerful mesh-free generative models for continuous stochastic processes, with critical applications in fields such as finance, physics, and biology. Previous state-of-the-art methods have relied on adversarial training, such as GANs, or on minimizing distance measures between processes using signature kernels. However, GANs suffer from issues like instability, mode collapse, and the need for specialized training techniques, while signature kernel-based methods require solving linear PDEs and backpropagating gradients through the solver, whose computational complexity scales quadratically with the discretization steps. In this paper, we identify a novel class of strictly proper scoring rules for comparing continuous Markov processes. This theoretical finding naturally leads to a novel approach called Finite Dimensional Matching (FDM) for training Neural SDEs. Our method leverages the Markov property of SDEs to provide a computationally efficient training objective. This scoring rule allows us to bypass the computational overhead associated with signature kernels and reduces the training complexity from $O(D^2)$ to $O(D)$ per epoch, where $D$ represents the number of discretization steps of the process. We demonstrate that FDM achieves superior performance, consistently outperforming existing methods in terms of both computational efficiency and generative quality.

## 1 Introduction

Stochastic differential equations (SDEs) are a modeling framework used to describe systems influenced by random forces, with applications spanning finance, physics, biology, and engineering. They incorporate stochastic terms to allow the modeling of complex systems under uncertainties.

A *neural stochastic differential equation* (Neural SDE) (Kidger et al., 2021; Issa et al., 2023; Tzen & Raginsky, 2019; Jia & Benson, 2019; Hodgkinson et al., 2021; Li et al., 2020; Morrill et al., 2020) is an SDE where neural networks parameterize the drift and diffusion terms. This model acts as a mesh-free generative model for time-series data and has shown a significant impact in financial applications (Arribas et al., 2021; Gierjatowicz et al., 2020; Choudhary et al., 2023; Hoglund et al., 2023).

Training Neural SDEs typically involves minimizing a distance measure between the distribution of generated paths and the distribution of observed data paths. State-of-the-art performance has been achieved using signature kernels to define a distance measure on path space (Issa et al., 2023). Although effective, this approach requires solving a linear partial differential equation (PDE) whose computational complexity scales quadratically with the discretization step, which becomes impractical for long time series. An alternative is training these models adversarially as Generative Adversarial Network (GAN) (Kidger et al., 2021). However, GAN-based training can be fraught with issues such as instability, mode collapse, and the need for specialized techniques.

---

[*]Most of the work was done during an internship at Cisco.

In this paper, we present a theoretical result that extends scoring rules for comparing distributions in finite-dimensional spaces to those for continuous Markov processes. This extension forms the basis of a novel algorithm, *Finite Dimensional Matching* (FDM), designed for training generative models of stochastic processes. FDM exploits the Markovian nature of SDEs by leveraging the two-time joint distributions of the process, providing an efficient training objective that bypasses the complexities of signature kernels. Notably, FDM reduces the computational complexity from $O(D^2)$ to $O(D)$ per training step, where $D$ represents the number of discretization steps.

The key contributions of this paper are as follows:

- Our main theorem shows that scoring rules to compare continuous Markov processes can be easily built upon scoring rules on finite-dimensional space.
- Our main theorem suggests an efficient training method, FDM, for Neural SDE.
- Our experiments show that FDM outperforms prior methods on multiple benchmarks.

The rest of the paper is organized as follows: Section 2 provides a review of the relevant literature. In Section 3, we present preliminary results that lay the foundation for our main contributions. Section 4 introduces our main theorem, which extends scoring rules for finite dimensions to continuous Markov processes and leads to the development of our novel *Finite Dimensional Matching* (FDM) algorithm. In section 5, we analyze the sample complexity and sensitivity of FDM. Section 6 details the experimental setup and results, demonstrating the superiority of FDM in terms of both computational efficiency and generative performance across several benchmark datasets. Finally, Section 7 concludes the paper by summarizing the contributions, limitations, and directions for future work.

## 2 RELATED WORK

We begin by reviewing prior applications of scoring rules in generative modeling followed by an exploration of the literature on training Neural SDEs.

### 2.1 SCORING RULES

Scoring rules offer a method to measure discrepancies between distributions (Gneiting & Raftery, 2007) and are especially appealing for generative modeling and have been employed in training various generative models (Bouchacourt et al., 2016; Gritsenko et al., 2020; Pacchiardi et al., 2024; Pacchiardi & Dutta, 2022; Issa et al., 2023; Bonnier & Oberhauser, 2024). Notably, Pacchiardi et al. (2024) apply scoring rules to discrete Markov chains, although their extension to continuous-time processes has not yet been explored. Issa et al. (2023) and Bonnier & Oberhauser (2024) construct scoring rules for continuous processes by utilizing signature kernels.

### 2.2 NEURAL SDES

Several methods have been proposed for training Neural SDEs as generative models, each differing in how they define the divergence or distance between distributions on path space. In Table 1, we compare different methods for training Neural SDEs, highlighting their divergence measures and the corresponding discriminator or training objectives. Our approach, **Finite Dimensional Matching** (FDM), introduces a novel scoring rule specifically designed for continuous Markov processes.

One method to train Neural SDE is the latent SDE model introduced by Li et al. (2020), which trains a Neural SDE using variational inference principles (Opper, 2019). In their framework, training involves optimizing the free energy that includes the Kullback-Leibler (KL) divergence between the original SDE and an auxiliary SDE. These two SDEs share the same diffusion term but have different drift terms. The KL divergence between their laws can be computed using Girsanov's change of measure theorem. However, the performance of latent SDEs is generally inferior to SDE-GANs due to lower model capacity (Kidger et al., 2021; Issa et al., 2023).

A prominent method is the SDE-GAN introduced by Kidger et al. (2021), which employs adversarial training to fit a Neural SDE, as in Wasserstein-GANs (Arjovsky et al., 2017). This approach relies on the 1-Wasserstein distance, with the discriminator parameterized by neural controlled differential equations (Kidger et al., 2020; Morrill et al., 2021). However, SDE-GANs are notoriously difficult

Table 1: Methods for training Neural SDEs. **SigKer** stands for signature kernel (Issa et al., 2023), **TruncSig** is for truncated signature (Bonnier & Oberhauser, 2024), **SDE-GAN** is proposed by Kidger et al. (2021), and **Latent SDE** is proposed by Li et al. (2020).

| Methods | Divergence or distance | Discriminator or training objective |
|---|---|---|
| **Latent SDE** | KL-divergence | Monte-Carlo simulation of free energy |
| **SDE-GAN** | 1-Wasserstein distance | Optimizing discriminator nets |
| **SigKer** | Signature kernel score | Solving Goursat PDEs |
| **TruncSig** | Truncated signature kernel score | Truncated approximation of signature |
| **FDM (Ours)** | A novel class of scoring rules for continuous Markov processes | Standard ERM of the expected scores |

to train due to their high sensitivity to hyperparameters. Another major challenge is the need for a Lipschitz discriminator, which requires techniques like weight clipping and gradient penalties to enforce this constraint (Kidger, 2022). Adversarial training for time-series generative models has also been explored in the context of discrete data (Ni et al., 2022; Yoon et al., 2019).

Another key contribution to training Neural SDEs is the signature kernel method (Issa et al., 2023; Bonnier & Oberhauser, 2024), which minimizes a distance measure based on signature kernels (Lee & Oberhauser, 2023) of paths. However, evaluation of the signature kernel requires solving Goursat partial differential equations (PDEs) and backpropagating gradients through the solver (Salvi et al., 2021). The computational complexity of solving Goursat PDEs scales quadratically with the number of discretization steps, which can be prohibitive for long time series data. Bonnier & Oberhauser (2024) approximates the signature kernel as inner products of truncated signature transforms, called truncated signature. However, the scoring rule based on truncated signature is not strictly proper and has $\mathcal{O}(d^N)$ memory complexity where $d$ is the number of features and $N$ is the truncation size.

The concurrent work of Snow & Krishnamurthy (2025) introduces a novel technique that leverages Wiener space cubature theory to bypass Monte Carlo simulations in Neural SDE training.

## 3 PRELIMINARIES

In this section, we set up the notations and introduce the following preliminary concepts: Neural SDEs, Markov processes, and scoring rules.

**Background and Notations** Let $\{\Omega, \mathcal{F}, \mathbb{P}\}$ be a probability space where $\Omega, \mathcal{F}, \mathbb{P}$ denote the sample space, sigma-algebra, and probability measure, respectively. For a random variable $\xi$, the function $\mathbb{P}_\xi = \mathbb{P} \circ \xi^{-1}$ is the induced measure on its range space. In particular, for a random process $X$, $\mathbb{P}_X$ denotes its law. We use the superscript $^\top$ for the transposition of a matrix or vector.

**Neural SDE** Let $B : [0, T] \times \Omega \to \mathbb{R}^{d_{noise}}$ be a Brownian motion on $\mathbb{R}^{d_{noise}}$, where $d_{noise} \in \mathbb{N}$. We define a Neural SDE as in Issa et al. (2023) and Kidger et al. (2021):

$$Z_0 = \xi^\theta(a), \ dZ_t = \mu^\theta(t, Z_t)dt + \sigma^\theta(t, Z_t)dW_t, \ X_t^\theta = A^\theta Z_t + b^\theta$$

where $a$ is sampled from a $d_{initial}$-dimensional standard Gaussian distribution,

$$\xi^\theta : \mathbb{R}^{d_{initial}} \to \mathbb{R}^{d_z}, \ \mu^\theta : [0, T] \times \mathbb{R}^{d_z} \to \mathbb{R}^{d_z}, \ \sigma^\theta : [0, T] \times \mathbb{R}^{d_z} \to \mathbb{R}^{d_z \times d_{noise}}$$

along with $A^\theta \in \mathbb{R}^{d_x \times d_z}$, $b^\theta \in \mathbb{R}^{d_x}$, are functions parameterized by neural networks, and $d_{initial}, d_x, d_z \in \mathbb{N}$. We assume additionally that $\mu^\theta$ and $\sigma^\theta$ are Lipschitz continuous in both arguments and $\xi^\theta(a)$ has finite second-order momentum. These conditions ensure that the SDE for $Z_t$ has a unique strong solution. Suppose $Y_t$ is the data process, we'd like to train the neural networks $\theta$ on data sampled from $\mathbb{P}_Y$ so that $\mathbb{P}_{X^\theta}$ matches $\mathbb{P}_Y$.

**Markov Process** We say a continuous process $X_t$ with filtration $\{\mathcal{F}_t\}$ is Markov if $X_u$ is independent of $\mathcal{F}_t$ for all $u \geq t$ given $X_t$ (Kallenberg, 2021). For an SDE of the form $dX_t = \mu(t, X_t) dt + \sigma(t, X_t) dB_t$, with the filtration generated by the Brownian motion $B_t$, $X_t$ is Markov as long as the SDE has a unique strong solution (Theorem 9.1, Mao (2007)).

**Scoring Rules** Given a measurable space $(\Omega_0, \mathcal{F}_0)$ and $\omega_0 \in \Omega_0$, a scoring rule (Gneiting & Raftery, 2007) $s(P, \omega_0)$ maps a probability measure $P$ on $\Omega_0$ and a sample $\omega_0$ to $\mathbb{R}$. The expected score is

defined as $S(P, Q) = \mathbb{E}_Q[s(P, \omega_0)] = \int_{\Omega_0} s(P, \omega_0) \, dQ(\omega_0)$, where $P$ is the predictive distribution and $Q$ is the true distribution. The scoring rule $s$ is said to be *proper* if the expected score satisfies $S(P, Q) \leq S(Q, Q)$. It is *strictly proper* if $S(P, Q) = S(Q, Q) \iff P = Q$. For example, let $k : \mathbb{R}^d \times \mathbb{R}^d \to \mathbb{R}$ be the RBF kernel defined as $k(x, y) = \exp\left(-\gamma \|x - y\|^2\right)$, where $\gamma > 0$ is a parameter that determines the width of the kernel, then $s(P, z) = \frac{1}{2}\mathbb{E}_{Z, Z' \sim P} k(Z, Z') - \mathbb{E}_{Z \sim P} k(Z, z)$ is a strictly proper scoring rule for distribution on $\mathbb{R}^d$ (Gneiting & Raftery, 2007).

Let $P^\theta$ be a distribution controlled by a generative model $\theta$, and let $Q$ be the true distribution accessed through data. Given a strictly proper scoring rule $s$, sufficient model capacity of $\theta$, and sufficient data points from $Q$, $P^\theta$ can be trained by maximizing $S(P^\theta, Q)$ over $\theta$, leading to $P^\theta = Q$. While many scoring rules for finite-dimensional spaces have been proposed, we lack strictly proper scoring rules for random processes that can be evaluated efficiently. In our main claim, we show that a strictly proper scoring rule for a two-time joint distribution, *i.e.*, the distributions $\{(X_{t_1}, X_{t_2}), \forall t_1, t_2 \in [0, T]\}$, for a random process $X$, can be converted into a strictly proper scoring rule for continuous Markov processes.

## 4 FINITE DIMENSIONAL MATCHING

In this section, we present our main theorem, which converts a scoring rule for a two-time joint distribution into a scoring rule for a Markov process. Specifically, if we have a scoring rule for $\Omega_0 = \mathbb{R}^{2d}$, then Theorem 2 allows us to convert it into a scoring rule for Markov processes $X, Y : [0, T] \to \mathbb{R}^d$, where $d \in \mathbb{N}$ and $T \in \mathbb{R}_{>0}$.

### 4.1 SCORING RULE FOR MARKOV PROCESS

In this section, we present our main theorem which shows that a strictly proper scoring rule for the two-time joint distributions can be converted to a scoring rule for two Markov processes. Let continuous Markov processes $X, Y$ on $[0, T]$ take values in a Polish space $\mathcal{E}$ endowed with its Borel $\sigma$-algebra. Let $s$ be any strictly proper scoring rule defined on $\mathcal{E} \times \mathcal{E}$. Let $S(P, Q) = \mathbb{E}_Q[s(P, \omega)] < \infty, \forall$ measures $P, Q$ on $\mathcal{E} \times \mathcal{E}$. We define the scoring rule $\bar{s}$ for continuous Markov processes as following:

**Definition 1.** $\bar{s}(\mathbb{P}_X, y) = \mathbb{E}_{(t_1, t_2) \sim U([0,T]^2)} s(\mathbb{P}_{(X_{t_1}, X_{t_2})}, (y_{t_1}, y_{t_2}))$, *where* $\mathbb{P}_{(X_{t_1}, X_{t_2})}$ *is the joint marginal distributions at times* $t_1, t_2$ *of* $X$, *and* $U([0, T]^2)$ *is the uniform distribution on* $[0, T]^2$.

Let $\bar{S}(\mathbb{P}_X, \mathbb{P}_Y) = \mathbb{E}_{y \sim \mathbb{P}_Y}[\bar{s}(\mathbb{P}_X, y)]$. Now we present our main claim, with its proofs deferred to the appendix.

**Theorem 2.** *If* $s$ *is a strictly proper scoring rule for distributions on* $\mathcal{E} \times \mathcal{E}$, $\bar{s}$ *is a strictly proper scoring rule for* $\mathcal{E}$-*valued continuous Markov processes on* $[0, T]$ *where* $T \in \mathbb{R}_{>0}$. *That is, for any* $\mathcal{E}$-*valued continuous Markov processes* $X, Y$ *with laws* $\mathbb{P}_X, \mathbb{P}_Y$, *respectively,* $\bar{S}(\mathbb{P}_X, \mathbb{P}_Y) \leq \bar{S}(\mathbb{P}_Y, \mathbb{P}_Y)$ *with equality achieved only if* $\mathbb{P}_X = \mathbb{P}_Y$.

In the appendix, we present a more generalized version of Theorem 2 that does not require the timestamps $t_1$ and $t_2$ to be sampled from $U([0, T]^2)$ in the definition of $\bar{s}$. Nonetheless, to maintain clarity and simplicity, we focus our discussion on the uniform sampling case in the main paper.

Suppose $X^\theta$ is a Markov process parameterized by neural net parameters $\theta$ with sufficient capacity. Therefore, maximizing $\bar{S}(\mathbb{P}_{X^\theta}, \mathbb{P}_Y) = \mathbb{E}_{Y \sim \mathbb{P}_Y}[\bar{s}(\mathbb{P}_{X^\theta}, Y)]$, which can be achieved by maximizing the corresponding empirical average, will result in $\mathbb{P}_{X^\theta} = \mathbb{P}_Y$.

We present a concrete example on the application of Theorem 2. Consider continuous Markov processes $X, Y$ on $[0, T]$ taking values in $\mathbb{R}^d$. Let $k : \mathbb{R}^{2d} \times \mathbb{R}^{2d} \to \mathbb{R}$ be the RBF kernel, recall that $s(P, z) = \frac{1}{2}\mathbb{E}_{Z, Z' \sim P} k(Z, Z') - \mathbb{E}_{Z \sim P} k(Z, z)$ is a strictly proper scoring rule for distribution on $\mathbb{R}^{2d}$ (Gneiting & Raftery, 2007). By Theorem 2,

$$\bar{s}(\mathbb{P}_X, y) = \mathbb{E}_{(t_1, t_2) \sim U([0,T]^2)} \left[ \frac{1}{2} \mathbb{E}_{X, X'} k([X_{t_1}, X_{t_2}], [X'_{t_1}, X'_{t_2}]) - \mathbb{E}_X k([X_{t_1}, X_{t_2}], [y_{t_1}, y_{t_2}]) \right]$$
(1)

is strictly proper, where $[\cdot, \cdot]$ is the concatenation of two vectors. $\bar{S}(\mathbb{P}_{X^\theta}, \mathbb{P}_Y) = \mathbb{E}_{Y \sim \mathbb{P}_Y}[\bar{s}(\mathbb{P}_{X^\theta}, Y)]$ can be estimated through empirical average and optimized efficiently.

### 4.2 FDM ALGORITHM

We consider an expected score $\bar{S}(\mathbb{P}_{X^\theta}, \mathbb{P}_Y) = \mathbb{E}_{Y \sim \mathbb{P}_Y}[\bar{s}(\mathbb{P}_{X^\theta}, Y)]$, which can be estimated using an empirical average $\hat{S}$. For example, an unbiased estimator of $\bar{S}$ for $\bar{s}$ defined in (1) can be constructed using batches of generated paths $\mathcal{B}_X = \{x^i\}_{i=1}^B$ and data paths $\mathcal{B}_Y = \{y^i\}_{i=1}^B$. For each $i$, independently sample two timestamps $t_i$ and $t'_i$. The empirical estimator is then given by:

$$\hat{S}(\mathcal{B}_X, \mathcal{B}_Y) = \frac{1}{2B(B-1)} \sum_{i \neq j} k\left([x^i_{t_j}, x^i_{t'_j}], [x^j_{t_j}, x^j_{t'_j}]\right) - \frac{1}{B^2} \sum_{i=1}^B \sum_{j=1}^B k\left([x^i_{t_j}, x^i_{t'_j}], [y^j_{t_j}, y^j_{t'_j}]\right).$$

Note that the above estimator $\hat{S}$ only requires each data path to be (potentially irregularly) observed at two distinct timestamps, and we can observe the $x^i$'s at any timestamps since they are generated by the Neural SDE model. Alternative empirical objectives are provided in the appendix.

In Algorithm 1, we present the concrete *finite dimensional matching* (FDM) algorithm derived from Theorem 2 to train a Neural SDE $X^\theta$.

---

**Algorithm 1:** Finite Dimensional Matching (FDM)

---

**Input:** Neural SDE $X^\theta$, data paths $\{y^i : i \in [N]\}$, strictly proper scoring rule $s$ , batch size $B$

1 **repeat**
2      Generate a batch of simulated paths $\mathcal{B}_X = \{x^i : i \in [B]\}$ using the Neural SDE model $\theta$;
3      Randomly sample a batch of data paths $\mathcal{B}_Y \subset \{y^i : i \in [N]\}$ with $|\mathcal{B}_Y| = B$ ;
4      Compute the empirical estimate $\hat{S}(\mathcal{B}_X, \mathcal{B}_Y)$ of $\bar{S}(\mathbb{P}_{X^\theta}, \mathbb{P}_Y)$;
5      Maximize $\hat{S}$ with respect to $\theta$ using an optimizer of the user's choice;
6 **until** *stopping criterion is met*;

---

## 5 THEORETICAL PROPERTIES

In this section, we investigate the sample complexity and sensitivity of the proposed scoring rules $\bar{s}$. All proofs are deffered to the appendix.

### 5.1 SAMPLE COMPLEXITY

We show that the sample complexity of the estimator $\hat{S}(\mathcal{B}_X, \mathcal{B}_Y)$ retains the classical sample complexity of a kernel-based scoring rule $s$ (Gretton et al., 2012). Let $k$ be a kernel associated with a *Reproducing Kernel Hilbert Space* (RKHS) and $s(P, z) = \frac{1}{2}\mathbb{E}_{Z,Z' \sim P}k(Z, Z') - \mathbb{E}_{Z \sim P}k(Z, z)$ be a strictly proper scoring rule. Recall that $\bar{s}(\mathbb{P}_X, y) = \mathbb{E}_{(t_1, t_2) \sim U([0,T]^2)} s(\mathbb{P}_{(X_{t_1}, X_{t_2})}, (y_{t_1}, y_{t_2}))$ and $\bar{S}(\mathbb{P}_X, \mathbb{P}_Y) = \mathbb{E}_{y \sim \mathbb{P}_Y}[\bar{s}(\mathbb{P}_X, y)]$.

**Theorem 3.** *Let $k(\cdot, \cdot)$ satisfy $0 \leq k(\cdot, \cdot) \leq K$ and the batch size $B \geq 2$. For any $\varepsilon > 0$,*

$$\mathbb{P}\left(|\hat{S} - \mathbb{E}[\hat{S}]| \geq \varepsilon\right) \leq 2 \exp\left(-\frac{8B\varepsilon^2}{47K^2}\right).$$

*Equivalently, with probability at least $1 - \delta$, the deviation of $\hat{S}$ from its expected value $\bar{S}(\mathbb{P}_X, \mathbb{P}_Y)$ is bounded as*

$$\left|\hat{S}(\mathcal{B}_X, \mathcal{B}_Y) - \bar{S}(\mathbb{P}_X, \mathbb{P}_Y)\right| \leq K\sqrt{\frac{47 \ln(2/\delta)}{8B}}.$$

$\hat{S}$ exhibits a sample complexity analogous to the classical sample complexity of kernel-based scoring rules $s$. In Section B of the appendix, we extend this analysis to an alternative estimator where all sample paths are evaluated at $n$ shared timestamps.

### 5.2 SENSITIVITY

Let $X$ be an Itô diffusion on $\mathbb{R}^d$, i.e., $dX_t = \mu(t, X_t)dt + \sigma(t, X_t)dB_t$. The following theorem shows how perturbations in $\mu$ and $\sigma$ affect the value of the scoring rule $\bar{s}(\mathbb{P}_X, y)$.

**Theorem 4.** *Let $X$ satisfy $dX_t = \mu(t, X_t)dt + \sigma(t, X_t)dB_t$ on $\mathbb{R}^d$. Let $\tilde{X}$ satisfy $d\tilde{X}_t = \tilde{\mu}(t, \tilde{X}_t)dt + \tilde{\sigma}(t, \tilde{X}_t)dB_t$ on $\mathbb{R}^d$ where $\forall t, x, \|\mu(t,x) - \tilde{\mu}(t,x)\|_2 \le \delta_\mu, \|\sigma(t,x) - \tilde{\sigma}(t,x)\|_2 \le \delta_\sigma$, and $\delta_\mu, \delta_\sigma$ are constants. Assume the scoring rule $s(P, z)$ is Lipschitz in terms of distribution $P$ with respect to the Wasserstein-2 distance. Assume both $X$ and $\tilde{X}$ have unique strong solutions and share the same initial conditions, then $|\bar{s}(\mathbb{P}_X, y) - \bar{s}(\mathbb{P}_{\tilde{X}}, y)| \le L_s C(\delta_\mu + \delta_\sigma)$, where $L_s$ is the Lipschitz constant of $s$, the constant $C$ depends on Lipschitz constants of $\mu$ and $\sigma$.*

If we allow different sampling distributions of $t_1, t_2$ than uniform as in the generalized main theorem Theorem 7 in the appendix, $C$ may also depend on the sampling distribution of $t_1, t_2$. This result provides a theoretical guarantee that small changes in the dynamics of the process result in changes to the scoring rule that are linear with respect to $\delta_\mu + \delta_\sigma$.

## 6 EXPERIMENTS

Table 2: Average KS test scores (**lower is better**) and chance of rejecting the null hypothesis (%) at 5%-significance level on marginals (**lower is better**) of metal prices, trained on paths evenly sampled at 64 timestamps.

| Dim | Model | $t = 6$ | $t = 19$ | $t = 32$ | $t = 44$ | $t = 57$ |
|---|---|---|---|---|---|---|
| SILVER | **SigKer** | .144, 23.9 | .134, 14.4 | .130, 11.5 | .126, 9.20 | .122, **7.96** |
| | **TruncSig** | .274, 97.9 | .277, 98.7 | .293, 99.3 | .304, 99.6 | .315, 99.6 |
| | **SDE-GAN** | .330, 70.0 | .647, 100. | .789, 100. | .813, 100. | .828, 100. |
| | **FDM (ours)** | **.118, 9.76** | **.114, 7.24** | **.112, 6.08** | **.114, 7.20** | **.117**, 8.16 |
| GOLD | **SigKer** | .129, 10.7 | .127, 9.40 | .128, 10.1 | .126, 9.80 | .123, **8.16** |
| | **TruncSig** | .255, 94.7 | .274, 98.4 | .298, 96.8 | .316, 99.8 | .330, 99.8 |
| | **SDE-GAN** | .244, 90.6 | .299, 94.8 | .318, 96.8 | .336, 96.8 | .352, 91.7 |
| | **FDM (ours)** | **.119, 9.32** | **.117, 8.00** | **.115, 7.34** | **.116, 7.66** | **.118**, 8.52 |

Table 3: Average KS test scores and chance of rejecting the null hypothesis (%) at 5%-significance level on marginals of U.S. stock indices, trained on paths evenly sampled at 64 timestamps. "DOLLAR", "USA30", "USA500", "USATECH", and "USSC2000" stand for US Dollar Index, USA 30 Index, USA 500 Index, USA 100 Technical Index, and US Small Cap 2000, respectively.

| Dim | Model | t=6 | t=19 | t=32 | t=44 | t=57 |
|---|---|---|---|---|---|---|
| DOLLAR | **SigKer** | .262, 76.4 | .316, 82.1 | .322, 83.7 | .314, 84.4 | .296, 83.9 |
| | **TruncSig** | .279, 98.3 | .303, 99.4 | .323, 99.7 | .339, 99.8 | .354, 99.9 |
| | **SDE-GAN** | .389, 93.0 | .544, 98.3 | .605, 99.5 | .599, 99.8 | .553, 99.8 |
| | **FDM (ours)** | **.143, 25.6** | **.151, 29.6** | **.153, 30.7** | **.155, 31.7** | **.156, 33.0** |
| USA30 | **SigKer** | .200, 56.5 | .239, 78.8 | .264, 91.8 | .279, 94.2 | .291, 93.7 |
| | **TruncSig** | .171, 51.5 | .194, 61.9 | .213, 70.9 | .228, 79.1 | .250, 89.5 |
| | **SDE-GAN** | .311, 80.9 | .402, 91.6 | .428, 90.6 | .550, 99.9 | .666, 100. |
| | **FDM (ours)** | **.132, 15.6** | **.123, 10.0** | **.124, 9.50** | **.124, 9.30** | **.121, 8.04** |
| USA500 | **SigKer** | .287, 86.8 | .350, 92.3 | .367, 94.0 | .365, 93.5 | .355, 93.1 |
| | **TruncSig** | .189, 57.7 | .204, 63.7 | .221, 71.3 | .231, 77.4 | .244, 86.1 |
| | **SDE-GAN** | .310, 97.0 | .448, 93.3 | .625, 100. | .713, 100. | .746, 100. |
| | **FDM (ours)** | **.122, 9.82** | **.117, 6.84** | **.117, 6.56** | **.117, 6.30** | **.118, 6.52** |
| USATECH | **SigKer** | .212, 80.6 | .240, 91.0 | .242, 90.9 | .239, 90.6 | .245, 92.2 |
| | **TruncSig** | .197, 74.1 | .227, 86.1 | .247, 91.3 | .265, 94.8 | .280, 97.6 |
| | **SDE-GAN** | .420, 99.9 | .786, 100. | .910, 100. | .950, 100. | .969, 100. |
| | **FDM (ours)** | **.123, 9.92** | **.119, 7.48** | **.118, 6.68** | **.118, 6.54** | **.118, 6.74** |
| USSC2000 | **SigKer** | .255, 70.1 | .312, 88.3 | .328, 94.5 | .325, 94.2 | .314, 93.6 |
| | **TruncSig** | .180, 46.5 | .199, 62.5 | .221, 78.8 | .233, 89.0 | .248, 96.4 |
| | **SDE-GAN** | .317, 75.4 | .572, 98.2 | .764, 100. | .843, 100. | .887, 100. |
| | **FDM (ours)** | **.134, 16.6** | **.126, 11.1** | **.122, 8.80** | **.124, 9.14** | **.122, 8.34** |

Table 4: Average KS test scores and the chance of rejecting the null hypothesis (%) at 5%-significance level on marginals for different currency pairs (EUR/USD and USD/JPY), trained on paths evenly sampled at 64 timestamps.

| Dim | Model | t=6 | t=19 | t=32 | t=44 | t=57 |
|---|---|---|---|---|---|---|
| EUR/USD | **SigKer** | .251, 66.5 | .293, 70.3 | .288, 66.4 | .271, 55.4 | .248, 38.5 |
| | **TruncSig** | .273, 97.9 | .313, 99.6 | .340, 99.8 | .354, 99.9 | .369, 99.9 |
| | **SDE-GAN** | .529, 89.2 | .665, 95.8 | .723, 96.0 | .754, 97.7 | .784, 99.8 |
| | **FDM (ours)** | **.125, 12.9** | **.113, 6.26** | **.109, 5.04** | **.110, 5.46** | **.111, 6.10** |
| USD/JPY | **SigKer** | .165, 34.3 | .189, 38.1 | .191, 34.4 | .188, 30.5 | .185, 29.3 |
| | **TruncSig** | .252, 87.8 | .291, 98.0 | .317, 99.4 | .334, 99.7 | .354, 99.9 |
| | **SDE-GAN** | .212, 73.4 | .267, 85.2 | .309, 88.6 | .359, 91.2 | .425, 92.8 |
| | **FDM (ours)** | **.120, 9.54** | **.111, 6.04** | **.110, 5.54** | **.111, 5.92** | **.111, 5.98** |

Table 5: Average KS test scores and the chance of rejecting the null hypothesis (%) at 5%-significance level on marginals of energy prices, trained on paths evenly sampled at 64 timestamps. We reserve the latest 20% data as test dataset and measure how well the model predicts into future. "BRENT", "DIESEL", "GAS", and "LIGHT" stand for U.S. Brent Crude Oil, Gas oil, Natural Gas, and U.S. Light Crude Oil, respectively.

| Dim | Model | t=6 | t=19 | t=32 | t=44 | t=57 |
|---|---|---|---|---|---|---|
| BRENT | **SigKer** | .284, 69.4 | .339, 70.7 | .343, 68.0 | .328, 65.4 | .302, 60.0 |
| | **TruncSig** | .254, 91.8 | .264, 95.4 | .273, 96.7 | .292, 98.1 | .303, 98.8 |
| | **SDE-GAN** | .487, 97.0 | .812, 100 | .929, 100 | .961, 100 | .981, 100 |
| | **FDM (ours)** | **.127, 12.9** | **.123, 10.2** | **.125, 11.5** | **.124, 11.6** | **.124, 11.4** |
| DIESEL | **SigKer** | .187, 47.1 | .218, 55.4 | .223, 49.6 | .222, 42.9 | .219, 38.1 |
| | **TruncSig** | .221, 81.9 | .244, 93.9 | .262, 97.9 | .274, 98.9 | .305, 99.6 |
| | **SDE-GAN** | .279, 77.9 | .522, 97.6 | .664, 99.8 | .735, 100 | .793, 100 |
| | **FDM (ours)** | **.122, 10.2** | **.117, 7.82** | **.117, 8.20** | **.123, 10.5** | **.123, 10.7** |
| GAS | **SigKer** | .244, 70.4 | .298, 77.1 | .305, 71.5 | .295, 69.1 | .273, 63.7 |
| | **TruncSig** | .244, 82.0 | .280, 93.4 | .301, 97.6 | .328, 99.4 | .342, 99.7 |
| | **SDE-GAN** | .337, 86.2 | .586, 99.8 | .717, 99.9 | .801, 100 | .877, 100 |
| | **FDM (ours)** | **.116, 7.52** | **.116, 7.36** | **.123, 11.7** | **.127, 14.4** | **.126, 13.7** |
| LIGHT | **SigKer** | .184, 60.1 | .200, 66.9 | .195, 59.1 | .186, 53.0 | .173, 43.6 |
| | **TruncSig** | .245, 91.7 | .261, 94.6 | .272, 96.4 | .292, 98.7 | .308, 99.5 |
| | **SDE-GAN** | .266, 74.8 | .403, 76.0 | .464, 85.7 | .604, 98.6 | .717, 99.9 |
| | **FDM (ours)** | **.121, 9.82** | **.122, 10.4** | **.131, 15.5** | **.131, 15.2** | **.132, 15.5** |

We evaluate our method, **FDM**[1], by comparing it to three existing methods for training Neural SDEs: the signature kernel method (**SigKer**, Issa et al. (2023)), the truncated signature method (**TruncSig**, Bonnier & Oberhauser (2024)), and **SDE-GAN** (Kidger et al., 2021). Our experiments are conducted across five real-world datasets: energy prices, bonds, metal prices, U.S. stock indices, and exchange rates, as well as one synthetic dataset, the Rough Bergomi model[2]. The real-world datasets are historical price data for variety of financial instruments. The rough Bergomi model is a widely used stochastic volatility model and has been extensively described in Issa et al. (2023). For all datasets, we model all features jointly with a single multi-dimensional Neural SDE.

Consistent with Issa et al. (2023), we use the Kolmogorov-Smirnov (KS) test to assess the marginal distributions for each dimension. Specifically, we compare a batch of generated paths against an unseen batch from the real data distribution and calculate the KS scores and the chance of rejecting the null hypothesis, which states that the two distributions are identical. This process is repeated

---

[1]code available at `https://github.com/Z-Jianxin/FDM`
[2]All real-world datasets are obtained from `https://www.dukascopy.com/swiss/english/marketwatch/historical/`

Table 6: Average KS test scores and chance of rejecting the null hypothesis (%) at 5%-significance level on marginals of bonds, trained on paths evenly sampled at 64 timestamps. We reserve the most latest 20% data as test dataset and measure how well the model predicts into future. "BUND", "UKGILT", and "USTBOND" stand for Euro Bund, UK Long Gilt, and US T-BOND, respectively.

| Dim | Model | t=6 | t=19 | t=32 | t=44 | t=57 |
|---|---|---|---|---|---|---|
| BUND | **SigKer** | .210, 49.0 | .244, 52.8 | .251, 54.7 | .252, 58.7 | .245, 54.3 |
| | **TruncSig** | .261, 95.0 | .296, 99.4 | .328, 99.7 | .350, 99.9 | .362, 99.9 |
| | **SDE-GAN** | .339, 97.8 | .582, 100 | .613, 99.3 | .764, 100 | .831, 100 |
| | **FDM (ours)** | **.119**, **10.2** | **.120**, **8.80** | **.124**, **8.66** | **.125**, **9.72** | **.118**, **8.20** |
| UKGILT | **SigKer** | .158, 26.0 | .183, 27.5 | .197, 30.5 | .201, 31.8 | .201, 32.0 |
| | **TruncSig** | .200, 67.8 | .244, 89.8 | .285, 98.9 | .312, 99.7 | .337, 99.8 |
| | **SDE-GAN** | .366, 84.3 | .640, 99.8 | .862, 100 | .902, 100 | .921, 100 |
| | **FDM (ours)** | **.127**, **12.6** | **.113**, **6.62** | **.109**, **5.28** | **.109**, **5.36** | **.109**, **5.70** |
| USTBOND | **SigKer** | .189, 40.6 | .207, 37.6 | .213, 35.7 | .213, 35.3 | .214, 36.2 |
| | **TruncSig** | .229, 73.5 | .248, 81.3 | .284, 95.0 | .310, 99.3 | .334, 99.8 |
| | **SDE-GAN** | .351, 84.4 | .688, 100 | .853, 100 | .903, 100 | .916, 100 |
| | **FDM (ours)** | **.137**, **17.9** | **.124**, **10.8** | **.115**, **6.60** | **.113**, **6.14** | **.111**, **5.64** |

Table 7: Average KS test scores and the chance of rejecting the null hypothesis (%) at 5%-significance level on marginals for different currency pairs (EUR/USD and USD/JPY), trained on paths evenly sampled at 256 timestamps.

| Dim | Model | $t = 25$ | $t = 76$ | $t = 128$ | $t = 179$ | $t = 230$ |
|---|---|---|---|---|---|---|
| EUR/USD | **SigKer** | .535, 100. | .535, 100. | .536, 100. | .546, 100. | .540, 100. |
| | **TruncSig** | .137, **18.9** | .184, 67.1 | .252, 99.6 | .290, 100. | .318, 100. |
| | **SDE-GAN** | **.134**, 21.6 | .411, 100 | .569, 100 | .548, 100 | .338, 99.9 |
| | **FDM (ours)** | .136, 23.0 | **.112**, **5.70** | **.123**, **12.7** | **.132**, **17.8** | **.141**, **26.6** |
| USD/JPY | **SigKer** | .535, 100. | .534, 100. | .535, 100. | .538, 100. | .541, 100. |
| | **TruncSig** | **.114**, **7.10** | .152, 28.3 | .199, 82.8 | .232, 97.9 | .242, 99.2 |
| | **SDE-GAN** | .201, 72.6 | .334, 99.9 | .407, 100. | .405, 100. | .338, 100. |
| | **FDM (ours)** | .124, 13.8 | **.112**, **6.30** | **.118**, **6.90** | **.122**, **9.00** | **.115**, **6.20** |

for all the test batches and we report the averaged KS scores and the chance of rejecting the null hypothesis across all the batches.

For all experiments, we use fully connected neural networks to parameterize the drift and diffusion terms, with hyperparameters and preprocessings suggested in Issa et al. (2023). We choose $s$ to be $s(P, z) = \frac{1}{2}\mathbb{E}_{Z,Z'\sim P}k(Z, Z') - \mathbb{E}_{Z\sim P}k(Z, z)$ where $k$ is the rbf kernel with unit kernel bandwidth. In particular, following Issa et al. (2023), we let our method and **TruncSig** train for 10000 steps, while **SDE-GAN** trains for 5000 steps and **SigKer** for 4000 steps, to normalize the training time. Despite the differences in training steps, our method remains the fastest in terms of wall-clock time. All models are trained and evaluated on a single NVIDIA H100 GPU.

For our experiments, we first follow Issa et al. (2023) to train and evaluate the models on three datasets—metal prices, stock indices, and exchange rates—using sequences with 64 timestamps and random train-test splits. This training and evaluation process is repeated with five different random seeds, and the average KS scores and rejection rates are reported in Tables 2, 3, and 4, respectively, with corresponding standard deviations provided in the appendix. For the energy price and bonds datasets, we reserve the latest 20% of the data for testing, evaluating the trained models via the KS test on generated sequences against unseen future sequences. These results are presented in Tables 5 and 6. We repeat the experiments on these two datasets five times, with standard deviations also reported in the appendix. Additionally, for the exchange rates dataset, we trained and tested the models using sequences with 256 and 1024 timestamps, reporting KS test results in Tables 7 and 8, and training time in Table 16 in the appendix. For the synthetic rough Bergomi model, we generated sequences with 64 timestamps across both 16 and 32 dimensions, with results reported

Table 8: Average KS test scores and the chance of rejecting the null hypothesis (%) at 5%-significance level on marginals for different currency pairs (EUR/USD and USD/JPY), trained on paths evenly sampled at 1024 timestamps. A thread limit error is encountered during the training of the **SigKer (Issa et al., 2023)**, which relies on a dedicated parallel PDE solver.

| Dim | Model | $t = 102$ | $t = 307$ | $t = 512$ | $t = 716$ | $t = 921$ |
|---|---|---|---|---|---|---|
| EUR/USD | **SigKer** | - | - | - | - | - |
| | **TruncSig** | .476, 100. | .718, 100. | .993, 100. | .996, 100. | .887, 100. |
| | **SDE-GAN** | .280, 98.4 | .818, 100. | .963, 100. | .846, 100. | .805, 100. |
| | **FDM (ours)** | **.117**, **11.1** | **.117**, **9.00** | **.138**, **25.1** | **.153**, **36.2** | **.191**, **66.5** |
| USD/JPY | **SigKer** | - | - | - | - | - |
| | **TruncSig** | .766, 100. | .743, 100. | .670, 100. | .998, 100. | 1.00, 100. |
| | **SDE-GAN** | .528, 100. | .291, 100. | .389, 100. | .530, 100. | .655, 100. |
| | **FDM (ours)** | **.138**, **20.9** | **.124**, **14.3** | **.150**, **32.1** | **.199**, **74.9** | **.260**, **97.9** |

Table 9: Average KS test scores and the chance of rejecting the null hypothesis (%) at 5%-significance level on marginals across all dimensions, trained on paths evenly sampled at 64 timestamps from a 16-dimension rough Bergomi model.

| Model | $t = 6$ | $t = 19$ | $t = 32$ | $t = 44$ | $t = 57$ |
|---|---|---|---|---|---|
| **SigKer** | **.112**, **6.60** | .118, **7.80** | .124, 10.8 | .132, 16.3 | .144, 25.5 |
| **TruncSig** | .450, 100. | .458, 100. | .462, 100. | .461, 100. | .460, 100. |
| **SDE-GAN** | .308, 99.8 | .374, 99.4 | .393, 99.5 | .406, 99.6 | .430, 99.7 |
| **FDM (ours)** | .113, 7.20 | **.116**, **7.80** | **.119**, **8.80** | **.124**, **11.8** | **.131**, **15.8** |

in Tables 9 and 10, where we report the average KS scores and the chance of rejecting the null hypothesis across different dimensions. We also compare the computational efficiency of the models in terms of training time for different dimensions of the Rough Bergomi model, with detailed results summarized in Table 17 in the appendix. We highlight the best-performing model across all tables.

We include qualitative studies in Figure 1, which compare the dynamics of joint distributions of real and generated data points for the metal price dataset. We compare the sample paths of the metal price dataset in Figure 2. Due to space constraints, we provide further qualitative studies comparing pairwise joint distributions and sample paths, along with tables comparing computational efficiency, in section G in the appendix. Our results demonstrate that our method outperforms competitors in an overwhelming majority of cases in terms of KS test results, qualitative results, and computational efficiency.

## 7 CONCLUSION, LIMITATIONS, AND FUTURE WORK

Our main theorem demonstrates that any strictly proper scoring rule for comparing distributions on finite dimensions can be extended to strictly proper scoring rules for comparing the laws of continuous Markov processes. This theorem naturally leads to the **FDM** algorithm for training Neural SDEs. We empirically show that **FDM** outperforms current state-of-the-art methods for training Neural SDEs, both in terms of generative quality and computational efficiency. However, the applicability of our main theorem is currently constrained by the assumptions of continuity and the Markov property. Although this lies beyond the scope of Neural SDEs, we provide a straightforward extension of the main theorem to Càdlàg Markov processes in the appendix. This extension broadens the applicability of **FDM** to a wider range of models, including jump processes. Furthermore, an intriguing direction for future work would be to relax the Markov assumptions, for instance, by incorporating hidden Markov models.

Table 10: Average KS test scores and the chance of rejecting the null hypothesis (%) at 5%-significance level on marginals across all dimensions, trained on paths evenly sampled at 64 timestamps from a 32-dimension rough Bergomi model. TruncSig runs out of GPU memory.

| Model | t=6 | t=19 | t=32 | t=44 | t=57 |
|---|---|---|---|---|---|
| **SigKer** | .120, 11.1 | .137, 18.5 | .149, 26.5 | .157, 35.3 | .168, 45.2 |
| **TruncSig** | - | - | - | - | - |
| **SDE-GAN** | .284, 99.8 | .288, 99.7 | .298, 99.8 | .311, 99.9 | .326, 100. |
| **FDM (ours)** | **.117**, **9.10** | **.119**, **10.2** | **.122**, **11.4** | **.124**, **13.0** | **.128**, **15.4** |

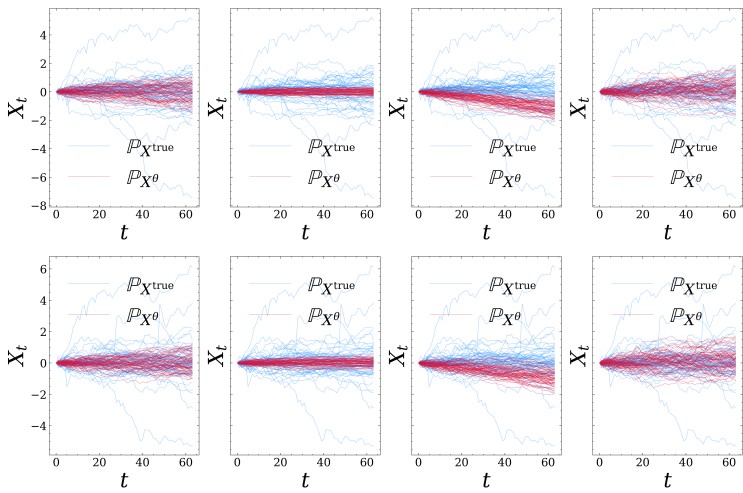

Figure 1: Blue points are real samples and orange points are generated by Neural SDEs. The dynamics of the joint distribution of gold and silver prices in the metal price data. Each row of plots corresponds to a method and each row corresponds to a timestamp. For each plot, the horizontal axis is the silver price and the vertical axis is the gold price.

Figure 2: Sample paths for silver (top) and gold (bottom) prices from the metal dataset. Blue lines represent real samples, while red lines represent those generated by Neural SDEs. From left to right, the plots correspond to signature kernels, truncated signature, SDE-GAN, and FDM, respectively. The horizontal axis represents time, and the vertical axis represents metal prices.

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
