$\mathcal{A}$ [3], let their transition kernels be $\mu_{u,v}^X(X_v, B) = \mathbb{P}(X_v \in B | X_u)$ and $\mu_{u,v}^Y(Y_v, B) = \mathbb{P}(Y_v \in B | Y_u)$ for $u, v \in \mathcal{T}$. For convenience, we use the kernel operations introduced in Chapter 3 of Kallenberg (2021). Let $B_1, B_2 \in \mathcal{A}$ and $t, u, v \in \mathcal{T}$, recall that $\mu_{t,u}^X \otimes \mu_{u,v}^X$ is given by

$$(\mu_{t,u}^X \otimes \mu_{u,v}^X)(x, B_1 \times B_2) = \int \mu_{t,u}^X(x, dz_1) \int \mu_{u,v}^X(z_1, dz_2) \mathbb{1}_{B_1 \times B_2}(z_1, z_2)$$

We need the following lemma to prove the main claim.

**Lemma 5.** *Let $\mathcal{T}$ be an index set. Let $X, Y$ be $\mathcal{E}$-valued Markov processes on $\mathcal{T}$. Then $X \overset{d}{=} Y$ $\iff \forall t_1, t_2 \in \mathcal{T}, (X_{t_1}, X_{t_2}) \overset{d}{=} (Y_{t_1}, Y_{t_2})$, where $\overset{d}{=}$ stands for equal in distribution.*

*Proof.* The $\implies$ direction is straightforward; we prove the other direction. Fix $t_1 \leq t_2 \in T$. Since $S$ is Borel, Theorem 8.5 of Kallenberg (2021) implies that the conditional distribution $\mu_{t_1, t_2}^X(z, \cdot) = \mu_{t_1, t_2}^Y(z, \cdot)$ for almost all $z$ under $\mathbb{P}_{X_{t_1}}$. By Proposition 11.2 of Kallenberg (2021), for any $t_0 \leq t_1 \cdots \leq t_n$ in $T$,

$$\begin{aligned} \mathbb{P}_{X_{t_0}, X_{t_1}, \ldots, X_{t_n}} &= \mathbb{P}_{X_{t_0}} \otimes \mu_{t_0, t_1}^X \otimes \cdots \otimes \mu_{t_{n-1}, t_n}^X \\ &= \mathbb{P}_{Y_{t_0}} \otimes \mu_{t_0, t_1}^Y \otimes \cdots \otimes \mu_{t_{n-1}, t_n}^Y \\ &= \mathbb{P}_{Y_{t_0}, Y_{t_1}, \ldots, Y_{t_n}}, \end{aligned}$$

*i.e.* $(X_{t_0}, X_{t_1}, \ldots, X_{t_n}) \overset{d}{=} (Y_{t_0}, Y_{t_1}, \ldots, Y_{t_n})$. Then $X \overset{d}{=} Y$ as their finite-dimensional distributions agree. $\qquad\square$

Recall that $\{\Omega, \mathcal{F}, \mathbb{P}\}$ is a probability space where $\Omega, \mathcal{F}, \mathbb{P}$ denote the sample space, sigma-algebra, and probability measure, respectively. Random processes $X, Y$ on $\mathcal{T} = [0, T]$ take values in a Polish space $\mathcal{E}$ endowed with its Borel $\sigma$-algebra $\mathcal{A}$. For a random variable $\xi$, the function $\mathbb{P}_\xi = \mathbb{P} \circ \xi^{-1}$ is the induced measure on its range space. In particular, for a random process $X$, $\mathbb{P}_X$ denotes its law. Let $s$ be any strictly proper scoring rule defined on $\mathcal{E} \times \mathcal{E}$ and $S(P, Q) = \mathbb{E}_Q[s(P, \omega)] < \infty, \forall$ measures $P, Q$ on $\mathcal{E} \times \mathcal{E}$ equipped with $\sigma$-algebra $\mathcal{A} \otimes \mathcal{A}$.

Here we present a more general version of Theorem 2 where $t_1$ and $t_2$ do not need to be uniformly sampled from $\mathcal{T}$. Let $\mu$ be the Lebesgue measure on $\mathcal{T}^2$. Let $\nu$ be a measure that is equivalent to $\mu$. That is, there exists the function $\lambda : \mathcal{T}^2 \to \mathbb{R}$ such that $\lambda(t_1, t_2) > 0$ $\mu$-a.e. and $\nu(A) = \int_A \lambda(t_1, t_2) d\mu$ for any measurable set $A$. We define the scoring rule $\bar{s}_\nu$ for continuous Markov processes with respect to the sampling measure $\nu$:

**Definition 6.** $\bar{s}_\nu(\mathbb{P}_X, y) = \mathbb{E}_{(t_1, t_2) \sim \nu} s(\mathbb{P}_{(X_{t_1}, X_{t_2})}, (y_{t_1}, y_{t_2}))$, *where $\mathbb{P}_{(X_{t_1}, X_{t_2})}$ is the joint marginal distributions at times $t_1, t_2$ of $X$.*

Let $\bar{S}_\nu(\mathbb{P}_X, \mathbb{P}_Y) = \mathbb{E}_{y \sim \mathbb{P}_Y}[\bar{s}_\nu(\mathbb{P}_X, y)]$. We present a generalized version of the main statement:

**Theorem 7.** *If $s$ is a strictly proper scoring rule for distributions on $\mathcal{E} \times \mathcal{E}$, $\bar{s}_\nu$ is a strictly proper scoring rule for $\mathcal{E}$-valued continuous Markov processes on $[0, T]$ where $T \in \mathbb{R}_{>0}$. That is, for any $\mathcal{E}$-valued continuous Markov processes $X, Y$ with laws $\mathbb{P}_X, \mathbb{P}_Y$, respectively, $\bar{S}_\nu(\mathbb{P}_X, \mathbb{P}_Y) \leq \bar{S}_\nu(\mathbb{P}_Y, \mathbb{P}_Y)$ with equality achieved only if $\mathbb{P}_X = \mathbb{P}_Y$.*

---

[3] $S$ is Borel isomorphic to a Borel set in $[0, 1]$. A Polish space with its Borel $\sigma$-algebra is Borel [p14, Kallenberg]

*Proof for Theorem 7.*

$$\bar{S}_\nu(\mathbb{P}_X, \mathbb{P}_Y) = \int \mathbb{E}_{(t_1,t_2)\sim\nu} s(\mathbb{P}_{(X_{t_1}, X_{t_2})}, (y_{t_1}, y_{t_2})) \quad \mathbb{P}_Y(dy)$$

$$= \mathbb{E}_{(t_1,t_2)\sim\nu} \int s(\mathbb{P}_{(X_{t_1}, X_{t_2})}, (y_{t_1}, y_{t_2})) \quad \mathbb{P}_Y(dy) \tag{2}$$

$$= \mathbb{E}_{(t_1,t_2)\sim\nu} \int s(\mathbb{P}_{(X_{t_1}, X_{t_2})}, (y_{t_1}, y_{t_2})) \quad \mathbb{P}_{(Y_{t_1}, Y_{t_2})}(d(y_{t_1}, y_{t_2})) \tag{3}$$

$$= \mathbb{E}_{(t_1,t_2)\sim\nu} S(\mathbb{P}_{(X_{t_1}, X_{t_2})}, \mathbb{P}_{(Y_{t_1}, Y_{t_2})}) \tag{4}$$

$$\leq \mathbb{E}_{(t_1,t_2)\sim\nu} S(\mathbb{P}_{(Y_{t_1}, Y_{t_2})}, \mathbb{P}_{(Y_{t_1}, Y_{t_2})}) \tag{5}$$

$$= \int \mathbb{E}_{(t_1,t_2)\sim\nu} s(\mathbb{P}_{(Y_{t_1}, Y_{t_2})}, (y_{t_1}, y_{t_2})) \quad \mathbb{P}_{(Y_{t_1}, Y_{t_2})}(d(y_{t_1}, y_{t_2})) \tag{6}$$

$$= \int \mathbb{E}_{(t_1,t_2)\sim\nu} s(\mathbb{P}_{(Y_{t_1}, Y_{t_2})}, (y_{t_1}, y_{t_2})) \quad \mathbb{P}_Y(dy) \tag{7}$$

$$= \bar{S}_\nu(\mathbb{P}_Y, \mathbb{P}_Y),$$

We apply Fubini's theorem for the (2) and use the substitution rule (Lemma 1.24, Kallenberg (2021)) (3). (4) and (5) follow from the definition of $S$ and the properness of the scoring rule $s$, respectively. Fubini's theorem and the substitution rule (Lemma 1.24, Kallenberg (2021)) are used again for the (6) and (7), respectively.

We then show strictness. Let $\bar{S}_\nu(\mathbb{P}_X, \mathbb{P}_Y) = \bar{S}_\nu(\mathbb{P}_Y, \mathbb{P}_Y)$. Then

$$\mathbb{E}_{(t_1,t_2)\sim\nu} S(\mathbb{P}_{(X_{t_1}, X_{t_2})}, \mathbb{P}_{(Y_{t_1}, Y_{t_2})}) = \mathbb{E}_{(t_1,t_2)\sim\nu} S(\mathbb{P}_{(Y_{t_1}, Y_{t_2})}, \mathbb{P}_{(Y_{t_1}, Y_{t_2})})$$

$$\iff \mathbb{E}_{(t_1,t_2)\sim\mu}\lambda(t_1,t_2) S(\mathbb{P}_{(X_{t_1}, X_{t_2})}, \mathbb{P}_{(Y_{t_1}, Y_{t_2})}) = \mathbb{E}_{(t_1,t_2)\sim\mu}\lambda(t_1,t_2) S(\mathbb{P}_{(Y_{t_1}, Y_{t_2})}, \mathbb{P}_{(Y_{t_1}, Y_{t_2})}).$$

So $S(\mathbb{P}_{(X_{t_1}, X_{t_2})}, \mathbb{P}_{(Y_{t_1}, Y_{t_2})}) = S(\mathbb{P}_{(Y_{t_1}, Y_{t_2})}, \mathbb{P}_{(Y_{t_1}, Y_{t_2})})$ $\mu$-*a.e.*. This implies $(X_{t_1}, X_{t_2}) \stackrel{d}{=} (Y_{t_1}, Y_{t_2})$ $\mu$-*a.e.*. Next, we show that this statement can be extended to all $(t_1, t_2)$.

Without loss of generality, let $(u_0, u'_0) \in [0, T]^2$ and $u_0 < u'_0$. We can inductively select $u_1, u_2, \ldots, u_n, \ldots$ and $u'_1, u'_2, \ldots, u'_n, \ldots$ such that $u_1 \in (u_0, \frac{u_0+u'_0}{2}]$, $u'_1 \in [\frac{u_0+u'_0}{2}, u'_0)$, $u_{n+1} \in (u_0, \frac{u_0+u_n}{2}]$, $u'_{n+1} \in [\frac{u'_n+u'_0}{2}, u'_0)$, and $(X_{u_n}, X_{u'_n}) \stackrel{d}{=} (Y_{u_n}, Y_{u'_n}) \forall n$. This is possible because $(u_0, \frac{u_0+u_n}{2}] \times [\frac{u'_n+u'_0}{2}, u'_0)$ has positive measure. Recall that $X$ and $Y$ are continuous processes. $(X_{u_n}, X_{u'_n})$ converges to $(X_{u_0}, X_{u'_0})$ and $(Y_{u_n}, Y_{u'_n})$ converges to $(Y_{u_0}, Y_{u'_0})$ almost surely as $u_n \to u_0$ and $u'_n \to u'_0$. Recall that $\mathcal{E} \times \mathcal{E}$ is also Polish. Then the convergence also holds in distribution and $(X_{u_0}, X_{u'_0}) \stackrel{d}{=} (Y_{u_0}, Y_{u'_0})$ (Lemma 5.2 and 5.7, Kallenberg (2021)).

By Lemma 5, $X \stackrel{d}{=} Y$. $\qquad\square$

Theorem 2 is a straightforward result of Theorem 7:

*Proof for Theorem 2.* Theorem 2 is a direct consequence of Theorem 7 by letting $\nu = \mu$. $\qquad\square$

## B PROOF OF SAMPLE COMPLEXITY

We'll use McDiarmid's inequality, due to McDiarmid (1989).

**Theorem 8.** *Let $X_1, X_2, \ldots, X_m$ be independent random variables taking values in a set $\mathcal{X}$. Let $f : \mathcal{X}^m \to \mathbb{R}$ be a function satisfying the bounded differences condition: for each $i \in \{1, \ldots, m\}$,*

$$\sup_{x_1,\ldots,x_m,x'_i} |f(x_1, \ldots, x_m) - f(x_1, \ldots, x_{i-1}, x'_i, x_{i+1}, \ldots, x_m)| \leq \Delta_i,$$

*where $\Delta_i \geq 0$ are constants. Then, for all $\varepsilon > 0$,*

$$\mathbb{P}\left(|f(X_1, \ldots, X_m) - \mathbb{E}[f(X_1, \ldots, X_m)]| \geq \varepsilon\right) \leq 2\exp\left(-\frac{2\varepsilon^2}{\sum_{i=1}^m \Delta_i^2}\right).$$

We state and prove the sample complexity results with a general sampling meausre $\nu$. Theorem 3 follows by letting $\nu = \mu$.

**Theorem 9.** *Let $\hat{S}(\mathcal{B}_X, \mathcal{B}_Y)$ be the empirical estimator defined as:*

$$\hat{S}(\mathcal{B}_X, \mathcal{B}_Y) = \frac{1}{2B(B-1)} \sum_{i \neq j} k\left([x_{t_j}^i, x_{t'_j}^i], [x_{t_j}^j, x_{t'_j}^j]\right) - \frac{1}{B^2} \sum_{i=1}^{B} \sum_{j=1}^{B} k\left([x_{t_j}^i, x_{t'_j}^i], [y_{t_j}^j, y_{t'_j}^j]\right),$$

*where:*

- $x^i$, $y^j$ *are independently sampled from $\mathbb{P}_X$ and $\mathbb{P}_Y$, respectively,*

- $t_j, t'_j$ *are independently sampled timestamp pairs, $\nu$ is a measure equivalent to the Lebesgue measure $\mu$ on $\mathcal{T}^2$*

- $k(\cdot, \cdot)$ *is a kernel function satisfying $0 \leq k(\cdot, \cdot) \leq K$.*

- $B \geq 2$.

*For any $\varepsilon > 0$,*

$$\mathbb{P}\left(|\hat{S} - \mathbb{E}[\hat{S}]| \geq \varepsilon\right) \leq 2 \exp\left(-\frac{8B\varepsilon^2}{47K^2}\right).$$

*Equivalently, with probability at least $1 - \delta$, the deviation of $\hat{S}$ from its expected value $\bar{S}(\mathbb{P}_{X^\theta}, \mathbb{P}_Y)$ is bounded as:*

$$\left|\hat{S}(\mathcal{B}_X, \mathcal{B}_Y) - \bar{S}(\mathbb{P}_{X^\theta}, \mathbb{P}_Y)\right| \leq K\sqrt{\frac{47 \ln(2/\delta)}{8B}}.$$

*Proof.* We will apply McDiarmid's inequality to the estimator $\hat{S}$. Recall that $\nu$ is a measure equivalent to the Lebesgue measure $\mu$ on $\mathcal{T}^2$. First, we verify the conditions of the inequality.

The estimator $\hat{S}$ depends on the independent variables:

- $x^i \in \mathcal{B}_X$: These are the generated paths. Changing a single $x^i$ while keeping other variables fixed changes $\hat{S}$ by at most $\frac{(2B-1)K}{2B(B-1)} + \frac{K}{B} \leq \frac{3K}{2B} + \frac{K}{B} = \frac{5K}{2B}$.

- $y^j \in \mathcal{B}_Y$: These are the data paths. Changing a single $y^j$ while keeping other variables fixed changes $\hat{S}$ by at most $\frac{K}{B}$.

- $t_j, t'_j \sim \nu$: These are the timestamps sampled from the measure $\nu$. Changing a single $t_j$ or $t'_j$ while keeping other variables fixed changes $\hat{S}$ by at most $\frac{3K}{2B}$.

Define the bounded differences:

$$\Delta_i = \begin{cases} \frac{3K}{2B}, & \text{for } i = 1, \ldots, 2B, \\ \frac{5K}{2B}, & \text{for } i = 2B+1, \ldots, 3B, \\ \frac{K}{B}, & \text{for } i = 3B+1, \ldots, 4B. \end{cases}$$

The sum of the squared bounded differences is:

$$\sum_{i=1}^{4B} \Delta_i^2 = 2B\left(\frac{3K}{2B}\right)^2 + B\left(\frac{5K}{2B}\right)^2 + B\left(\frac{K}{B}\right)^2 = \frac{47K^2}{4B}.$$

By McDiarmid's inequality, for any $\varepsilon > 0$:

$$\mathbb{P}\left(|\hat{S} - \mathbb{E}[\hat{S}]| \geq \varepsilon\right) \leq 2 \exp\left(-\frac{2\varepsilon^2}{\sum_{i=1}^{4B} \Delta_i^2}\right) = 2 \exp\left(-\frac{8B\varepsilon^2}{47K^2}\right).$$

$\square$

We analyze the sample complexity for a different estimator where all sample paths are observed at $n$ shared timestamps.

**Theorem 10.** *Let $\hat{S}$ be the empirical estimator defined as:*

$$\hat{S} = \frac{1}{n} \sum_{r=1}^{n} \left[ \frac{1}{2m(m-1)} \sum_{\substack{i,j=1 \\ i \neq j}}^{m} k\left([x_{t_r}^i, x_{t_r'}^i], [x_{t_r}^j, x_{t_r'}^j]\right) - \frac{1}{m^2} \sum_{i=1}^{m} \sum_{j=1}^{m} k\left([x_{t_r}^i, x_{t_r'}^i], [y_{t_r}^j, y_{t_r'}^j]\right) \right],$$

*where:*

- $x^i$, $y^j$ *are independently sampled from $\mathbb{P}_X$ and $\mathbb{P}_Y$, respectively,*

- $t_r, t_r' \sim \nu$ *are independently sampled timestamp pairs, $\nu$ is a measure equivalent to the Lebesgue measure $\mu$ on $\mathcal{T}^2$*

- $k(\cdot, \cdot)$ *is a kernel function satisfying $0 \leq k(\cdot, \cdot) \leq K$.*

- $m \geq 2$.

*For any $\varepsilon > 0$,*

$$\mathbb{P}\left(|\hat{S} - \mathbb{E}[\hat{S}]| \geq \varepsilon\right) \leq 2 \exp\left(-\frac{8mn\varepsilon^2}{K^2(29n + 18m)}\right).$$

*where $\mathbb{E}[\hat{S}] = \bar{S}_\nu(\mathbb{P}_X, \mathbb{P}_Y) = \mathbb{E}_{Y \sim \mathbb{P}_Y}[\bar{s}_\nu(\mathbb{P}_X, Y)]$.*

*Proof.* We will apply McDiarmid's inequality to the estimator $\hat{S}$. First, we verify the conditions of the inequality.

The estimator $\hat{S}$ depends on the independent variables:

- Generated paths $x^i$: Changing a single $x^i$ while keeping other variables fixed changes $\hat{S}$ by at most $\frac{(2m-1)K}{2m(m-1)} + \frac{K}{m} \leq \frac{3K}{2m} + \frac{K}{m} = \frac{5K}{2m}$.

- Data paths $y^j$: Changing a single $y^j$ while keeping other variables fixed changes $\hat{S}$ by at most $\frac{K}{m}$.

- Timestamp pairs $(t_r, t_r') \sim \nu$: Changing a single timestamp $t_r$ or $t_r'$ while keeping other variables fixed changes $\hat{S}$ by at most $\frac{3K}{2n}$.

Define the bounded differences:

$$\Delta_i = \begin{cases} \frac{5K}{2m}, & \text{for } i = 1, \ldots, m, \\ \frac{K}{m}, & \text{for } i = m+1, \ldots, 2m, \\ \frac{3K}{2n}, & \text{for } i = 2m+1, \ldots, 2m+2n. \end{cases}$$

The sum of the squared bounded differences is:

$$\sum_{i=1}^{2m+2n} \Delta_i^2 = m\left(\frac{5K}{2m}\right)^2 + m\left(\frac{K}{m}\right)^2 + 2n\left(\frac{3K}{2n}\right)^2.$$

Simplifying each term:

$$\sum_{i=1}^{2m+2n} \Delta_i^2 = \frac{25K^2}{4m} + \frac{K^2}{m} + \frac{9K^2}{2n} = K^2\left(\frac{29}{4m} + \frac{9}{2n}\right).$$

By McDiarmid's inequality, for any $\varepsilon > 0$:

$$\mathbb{P}\left(|\hat{S} - \mathbb{E}[\hat{S}]| \geq \varepsilon\right) \leq 2 \exp\left(-\frac{2\varepsilon^2}{\sum_{i=1}^{2m+2n} \Delta_i^2}\right) = 2 \exp\left(-\frac{8mn\varepsilon^2}{K^2(29n + 18m)}\right).$$

Equivalently, with probability at least $1 - \delta$:

$$\big| \hat{S} - \mathbb{E}[\hat{S}] \big| \leq K\sqrt{\frac{(29n + 18m)}{8mn}\ln\left(\frac{2}{\delta}\right)}.$$

$\square$

The sample complexity bound established in Theorem 10 demonstrates that the generalization error of the kernel-based scoring rule $\bar{s}$ is influenced not only by the number of sample paths $m$, as is the case for traditional scoring rules where the complexity depends on $m$ through $1/\sqrt{m}$, but also by the sampling frequency $n$ of the timestamp pairs.

## C  PROOF OF SENSITIVITY

We first prove the following lemma to bound the difference process $\Delta_t = X_t - \tilde{X}_t$.

**Lemma 11.** *Let $X$ satisfy $dX_t = \mu(t, X_t)dt + \sigma(t, X_t)dB_t$ on $\mathbb{R}^d$ for $t \in [0, T]$. Let $\tilde{X}$ satisfy $d\tilde{X}_t = \tilde{\mu}(t, \tilde{X}_t)dt + \tilde{\sigma}(t, \tilde{X}_t)dB_t$ on $\mathbb{R}^d$ for $t \in [0, T]$ where $\forall t, x, \|\mu(t, x) - \tilde{\mu}(t, x)\|_2 \leq \delta_\mu$, $\|\sigma(t, x) - \tilde{\sigma}(t, x)\|_2 \leq \delta_\sigma$, and $\delta_\mu$, $\delta_\sigma$ are constants. Assume $X$ and $\tilde{X}$ share the same initial conditons. Assume both $X$ and $\tilde{X}$ have unique strong solutions so $\mu$ and $\sigma$ are Lipschitz with constant $L_\mu$ and $L_\sigma$, respectively. Then $\forall t \in [0, T]$,*

$$\mathbb{E}\|\Delta_t\|_2^2 \leq \left(\frac{2D^2}{C} + 1\right)(\delta_\mu + \delta_\sigma)^2 t e^{\frac{3C}{2}t},$$

*where $D = \max(1, L_\sigma)$ and $C = 2L_\mu + L_\sigma^2$.*

*Proof.* Apply Ito's lemma,

$$d\|\Delta_t\|_2^2 = 2\Big\langle\Delta_t, \mu(t, X_t) - \tilde{\mu}(t, \tilde{X}_t)\Big\rangle dt + \Big\|\sigma(t, X_t) - \tilde{\sigma}(t, \tilde{X}_t)\Big\|_2^2 dt$$
$$+ 2\Big\langle\Delta_t, \sigma(t, X_t) - \tilde{\sigma}(t, \tilde{X}_t)\Big\rangle dB_t$$

Hence,

$$\mathbb{E}\|\Delta_t\|_2^2 = 2\mathbb{E}\left[\int_0^t \Big\langle\Delta_s, \mu(s, X_s) - \tilde{\mu}(s, \tilde{X}_s)\Big\rangle ds + \int_0^t \Big\|\sigma(s, X_s) - \tilde{\sigma}(s, \tilde{X}_s)\Big\|_2^2 ds\right] \quad (8)$$

Using the Lipsthitz property of $\mu$ and the bounded difference between $\mu$ and $\tilde{\mu}$,

$$\Big\langle\Delta_s, \mu(s, X_s) - \tilde{\mu}(s, \tilde{X}_s)\Big\rangle \leq \|\Delta_s\|_2\Big\|\mu(s, X_s) - \mu(s, \tilde{X}_s) + \mu(s, \tilde{X}_s) - \tilde{\mu}(s, \tilde{X}_s)\Big\|_2$$
$$\leq \|\Delta_s\|_2\Big(\Big\|\mu(s, X_s) - \mu(s, \tilde{X}_s)\Big\|_2 + \Big\|\mu(s, \tilde{X}_s) - \tilde{\mu}(s, \tilde{X}_s)\Big\|_2\Big)$$
$$\leq \|\Delta_s\|_2(L_\mu\|\Delta_s\|_2 + \delta_\mu) \quad (9)$$

Apply the plus-minus trick again, we have

$$\Big\|\sigma(s, X_s) - \tilde{\sigma}(s, \tilde{X}_s)\Big\|_2^2 \leq (L_\sigma\|\Delta_s\|_2 + \delta_\sigma)^2 \quad (10)$$

Substitute (9) and (10) back to equation (8) and apply Cauchy-Schwarz on $\mathbb{E}\|\Delta_t\|_2$,

$$\mathbb{E}\|\Delta_t\|_2^2 \leq \int_0^t (2L_\mu + L_\sigma^2)\mathbb{E}\|\Delta_s\|_2^2 + 2(\delta_\mu + L_\sigma\delta_\sigma)\mathbb{E}\|\Delta_s\|_2 + \delta_\sigma^2 ds$$
$$\leq \int_0^t (2L_\mu + L_\sigma^2)\mathbb{E}\|\Delta_s\|_2^2 + 2(\delta_\mu + L_\sigma\delta_\sigma)\sqrt{\mathbb{E}\|\Delta_s\|_2^2} + \delta_\sigma^2 ds$$

Let $f(t) = \mathbb{E}\|\Delta_t\|_2^2$, $C = 2L_\mu + L_\sigma^2$ and $D = \max(1, L_\sigma)$. Then,

$$f(t) \le \int_0^t C f(s) + 2D(\delta_\mu + \delta_\sigma)\sqrt{f(s)} + \delta_\sigma^2 ds$$

Recall the inequality, $\forall \epsilon > 0, a \in \mathbb{R}, a\sqrt{f(s)} \le \frac{a^2}{2\epsilon} + \frac{\epsilon}{2} f(s)$. Let $\epsilon = C$, we have,

$$f(t) \le \int_0^t \frac{3C}{2} f(s) + \frac{2D^2(\delta_\mu + \delta_\sigma)^2}{C} + \delta_\sigma^2 ds = \frac{3C}{2}\int_0^t f(s) ds + \left(\frac{2D^2}{C} + 1\right)(\delta_\mu + \delta_\sigma)^2 t.$$

Apply Gronwall's inequality,

$$\mathbb{E}\|\Delta_t\|_2^2 = f(t) \le \left(\frac{2D^2}{C} + 1\right)(\delta_\mu + \delta_\sigma)^2 t e^{\frac{3Ct}{2}}$$

$\square$

Now we're ready to prove Theorem 4.

*Proof of Theorem 4.* We again work with the general scoring rule $\bar{s}_\nu$ defined in Definition 6 where the sampling measure $\nu$ can be any measure equivalent to the Lesbegue measure $\mu$.

Recall that the scoring rule is Lipshtiz with constant $L_s$, with respect to the Wasserstein-2 distance $W_2$, so $\forall z$, and measures $P, P'$, $|s(P, z) - s(P', z)| \le L_s W_2(P, P')$.

Hence, for any realization $y$,

$$\left|\bar{s}_\nu(\mathbb{P}(X, y)) - \bar{s}_\nu(\mathbb{P}(\tilde{X}, y))\right| = \left|\mathbb{E}_{t_1, t_2 \sim \nu}\left[s(\mathbb{P}_{(X_{t_1}, X_{t_2})}, y) - s(\mathbb{P}_{(\tilde{X}_{t_1}, \tilde{X}_{t_2})}, y)\right]\right|$$

$$\le \mathbb{E}_{t_1, t_2 \sim \nu}\left|s(\mathbb{P}_{(X_{t_1}, X_{t_2})}, y) - s(\mathbb{P}_{(\tilde{X}_{t_1}, \tilde{X}_{t_2})}, y)\right|$$

$$\le L_s \mathbb{E}_{t_1, t_2 \sim \nu}\left[W_2\left(\mathbb{P}_{(X_{t_1}, X_{t_2})}, \mathbb{P}_{(\tilde{X}_{t_1}, \tilde{X}_{t_2})}\right)\right]$$

Let $\Gamma(\cdot, \cdot)$ be the couplings of two measures. Then,

$$W_2^2\left(\mathbb{P}_{(X_{t_1}, X_{t_2})}, \mathbb{P}_{(\tilde{X}_{t_1}, \tilde{X}_{t_2})}\right)$$

$$= \inf_{\gamma \in \Gamma\left(\mathbb{P}_{(X_{t_1}, X_{t_2})}, \mathbb{P}_{(\tilde{X}_{t_1}, \tilde{X}_{t_2})}\right)} \mathbb{E}_\gamma \left\|[X_{t_1}, X_{t_2}] - [\tilde{X}_{t_1}, \tilde{X}_{t_2}]\right\|_2^2 \tag{11}$$

$$= \inf_{\gamma \in \Gamma\left(\mathbb{P}_{(X_{t_1}, X_{t_2})}, \mathbb{P}_{(\tilde{X}_{t_1}, \tilde{X}_{t_2})}\right)} \mathbb{E}_\gamma \left\|X_{t_1} - \tilde{X}_{t_1}\right\|_2^2 + \left\|X_{t_2} - \tilde{X}_{t_2}\right\|_2^2$$

$$\le \mathbb{E}\|\Delta_{t_1}\|_2^2 + \mathbb{E}\|\Delta_{t_2}\|_2^2$$

$$\le \left(\frac{2D^2}{C} + 1\right)(\delta_\mu + \delta_\sigma)^2 (t_1 e^{\frac{3Ct_1}{2}} + t_2 e^{\frac{3Ct_2}{2}}), \tag{12}$$

where (11) follows the definition of $W_2$, (12) follows Lemma 11, and $C, D$ are defined in Lemma 11.

Finally,

$$\left|\bar{s}_\nu(\mathbb{P}(X, y)) - \bar{s}_\nu(\mathbb{P}(\tilde{X}, y))\right| \le L_s \mathbb{E}_{t_1, t_2 \sim \nu}\left[W_2\left(\mathbb{P}_{(X_{t_1}, X_{t_2})}, \mathbb{P}_{(\tilde{X}_{t_1}, \tilde{X}_{t_2})}\right)\right]$$

$$\le L_s \mathbb{E}_{t_1, t_2 \sim \nu}\left[\sqrt{t_1 e^{\frac{3Ct_1}{2}} + t_2 e^{\frac{3Ct_2}{2}}}\right]\sqrt{1 + \frac{2D^2}{C}}(\delta_\mu + \delta_\sigma)$$

The proof is then concluded by renaming the constants. $\square$

# D    EXTENSION TO CÀDLÀG MARKOV PROCESS

We show that the proof can be extended to Càdlàg Markov processes, where the paths $t \mapsto X_t$ are right-continuous with left limits everywhere, with probability one. Although this extension goes beyond the scope of Neural SDEs, such processes encompass a wide range of applications. Consider Càdlàg Markov processes $X, Y$ on $\mathcal{T}' = [0, T)$ that take values in a Polish space $\mathcal{E}$ endowed with its Borel $\sigma$-algebra $\mathcal{A}$. Let $s$ be any strictly proper scoring rule defined on $\mathcal{E} \times \mathcal{E}$, and let $S(P, Q) = \mathbb{E}_Q[s(P, \omega)] < \infty$ for all measures $P, Q$ on $\mathcal{E} \times \mathcal{E}$ equipped with $\sigma$-algebra $\mathcal{A} \otimes \mathcal{A}$.

We generalize Theorem 7 to Càdlàg Markov processes. Let $\mu$ denote the Lebesgue measure on $\mathcal{T}' \times \mathcal{T}'$, and let $\nu$ be a measure equivalent to $\mu$. We define $\bar{s}_\nu$ as in Definition 6. Let $\bar{S}_\nu(\mathbb{P}_X, \mathbb{P}_Y) = \mathbb{E}_{y \sim \mathbb{P}_Y}[\bar{s}_\nu(\mathbb{P}_X, y)]$. The main statement is presented below in its Càdlàg form:

**Theorem 12.** *If $s$ is a strictly proper scoring rule for distributions on $\mathcal{E} \times \mathcal{E}$, $\bar{s}_\nu$ is a strictly proper scoring rule for $\mathcal{E}$-valued Càdlàg Markov processes on $[0, T)$ where $T \in \mathbb{R}_{>0}$. That is, for any $\mathcal{E}$-valued Càdlàg Markov processes $X, Y$ with laws $\mathbb{P}_X, \mathbb{P}_Y$, respectively, $\bar{S}_\nu(\mathbb{P}_X, \mathbb{P}_Y) \leq \bar{S}_\nu(\mathbb{P}_Y, \mathbb{P}_Y)$ with equality achieved only if $\mathbb{P}_X = \mathbb{P}_Y$.*

*Proof.* Following the proof of theorem 7, we can show that $(X_{t_1}, X_{t_2}) \stackrel{d}{=} (Y_{t_1}, Y_{t_2})$ $\mu$-a.e.. We show that this statement can be extended to all $(t_1, t_2) \in \mathcal{T}' \times \mathcal{T}'$ using the right continuity.

Without loss of generality, let $(u_0, u'_0) \in [0, T)^2$ and $u_0 < u'_0$. We can inductively select $u_1, u_2, \ldots, u_n, \ldots$ and $u'_1, u'_2, \ldots, u'_n, \ldots$ such that $u_1 \in (u_0, \frac{u_0 + u'_0}{2}]$, $u'_1 \in [u'_0, T)$, $u_{n+1} \in (u_0, \frac{u_0 + u_n}{2}]$, $u'_{n+1} \in [u'_0, \frac{u'_n + u'_0}{2})$, and $(X_{u_n}, X_{u'_n}) \stackrel{d}{=} (Y_{u_n}, Y_{u'_n}) \forall n$. This is possible because $(u_0, \frac{u_0 + u_n}{2}] \times [u'_0, \frac{u'_n + u'_0}{2})$ has positive measure. Recall that $X$ and $Y$ are Càdlàg processes. $(X_{u_n}, X_{u'_n})$ converges to $(X_{u_0}, X_{u'_0})$ and $(Y_{u_n}, Y_{u'_n})$ converges to $(Y_{u_0}, Y_{u'_0})$ almost surely as $u_n \to u_0$ and $u'_n \to u'_0$. Recall that $\mathcal{E} \times \mathcal{E}$ is also Polish. Then the convergence also holds in distribution and $(X_{u_0}, X_{u'_0}) \stackrel{d}{=} (Y_{u_0}, Y_{u'_0})$ (Lemma 5.2 and 5.7, Kallenberg (2021)). $\square$

# E    COMPUTATIONAL EFFICIENCY

In this section, we clarify and explain the reduction in computational complexity achieved by our proposed method.

The $O(D^2)$ complexity arises from the previous state-of-the-art Neural SDE training method proposed in Issa et al. (2023), which involves solving a partial differential equation (PDE):

$$f(s, t) = 1 + \int_0^s \int_0^t f(u, v) \langle dx_u, dy_v \rangle_1 dv du,$$

as shown in Equation (2) of their paper. Backpropagation through the PDE solver introduces significant computational cost.

To approximate the double integral numerically, a rectangular rule with $D$ discretization steps is typically employed:

$$\int_0^T \int_0^T f(u, v) \langle dx_u, dy_v \rangle_1 dv du \approx \sum_{i=1}^D \sum_{j=1}^D f(u_i, v_j) \langle dx_{u_i}, dy_{v_j} \rangle \Delta u \Delta v,$$

where $\Delta u = T/D$, $\Delta v = T/D$, and $u_i = i\Delta u$, $v_j = j\Delta v$ for $i, j = 1, \ldots, D$. This double sum results in $O(D^2)$ complexity.

Furthermore, their method involves a double sum over the batch size $B$ in the objective function (Equation (4) in their paper). Our $B$ corresponds to their $m$, and the double integral appears in their $k_{sig}$ term. Consequently, their overall complexity is $O(D^2 B^2)$.

Our proposed method reduces the complexity from $O(D^2)$ to $O(D)$, or from $O(D^2 B^2)$ to $O(D B^2)$ when considering the batch size. This improvement is achieved because our approach eliminates the need to solve the PDE with the double integral, avoiding the computationally expensive operations required by the previous method.

## F   ALTERNATIVE EMPIRICAL OBJECTIVES

**Empirical Objective: Multiple Observations Concatenated.** An unbiased estimator can be constructed using batches of generated paths $\mathcal{B}_X = \{x^i\}_{i=1}^{B}$ and data paths $\mathcal{B}_Y = \{y^i\}_{i=1}^{B}$, where each path is observed at more than two timestamps. Suppose for each $i$, we select multiple (potentially irregular) observations at timestamps $t_i^1, t_i^2, \ldots, t_i^N$ where $N$ itself can be random or a tuning parameter. We concatenate these multiple observations to form vectors $[x_{t_i^1}^i, x_{t_i^2}^i, \ldots, x_{t_i^N}^i]$. The empirical estimator is then given by:

$$
\begin{aligned}
\hat{S}_1(\mathcal{B}_X, \mathcal{B}_Y) = & \frac{1}{2B(B-1)} \sum_{i \neq j} k\left([x_{t_j^1}^i, \ldots, x_{t_j^N}^i], \ [x_{t_j^1}^j, \ldots, x_{t_j^N}^j]\right) \\
& - \frac{1}{B^2} \sum_{i=1}^{B} \sum_{j=1}^{B} k\left([x_{t_j^1}^i, \ldots, x_{t_j^N}^i], \ [y_{t_j^1}^j, \ldots, y_{t_j^N}^j]\right).
\end{aligned}
$$

**Empirical Objective: Adjacent Timestamps as IID Samples.** Alternatively, we consider every pair of adjacent timestamps as independent and identically distributed (i.i.d.) samples. Suppose each data path is observed at timestamps $t_i^1 < t_i^2 < \ldots < t_i^M$. For each pair of adjacent timestamps $(t_i^m, t_i^{m+1})$, we treat the (potentially irregular) observations as i.i.d. samples. The empirical estimator is then:

$$
\begin{aligned}
\hat{S}_2(\mathcal{B}_X, \mathcal{B}_Y) = & \frac{1}{2B(M-1)(B-1)} \sum_{m=1}^{M-1} \sum_{i \neq j} k\left([x_{t_j^m}^i, x_{t_j^{m+1}}^i], \ [x_{t_j^m}^j, x_{t_j^{m+1}}^j]\right) \\
& - \frac{1}{B^2(M-1)} \sum_{m=1}^{M-1} \sum_{i=1}^{B} \sum_{j=1}^{B} k\left([x_{t_j^m}^i, x_{t_j^{m+1}}^i], \ [y_{t_j^m}^j, y_{t_j^{m+1}}^j]\right).
\end{aligned}
$$

Note that both estimators only require each data path to be observed at multiple timestamps, which can be irregular and path-dependent. All three empirical objectives, including the one presented in the main paper, perform similarly well in our preliminary experiments.

## G   ADDITIONAL EXPERIMENTAL RESULTS

The standard deviations on metal prices, stock indices, exchange rates, energy price, and bonds datasets are reported in Tables 11, 12, 13, 14, 15, respectively.

We evaluate the computational efficiency of the models by comparing their training times across different numbers of timestamps for the exchange rates dataset, with the detailed results presented in Table 16. Additionally, we assess the training times for various dimensions of the Rough Bergomi model, with the corresponding results summarized in Table 17.

Table 11: Standard deviations of average KS test scores and chance of rejecting the null hypothesis (%) at 5%-significance level on marginals of metal prices, trained on paths evenly sampled at 64 timestamps.

| Dim | Model | $t = 6$ | $t = 19$ | $t = 32$ | $t = 44$ | $t = 57$ |
|---|---|---|---|---|---|---|
| SILVER | **SigKer** | .020, 20.0 | .013, 9.26 | .008, 3.96 | .006, 2.47 | .006, 2.37 |
| | **TruncSig** | .080, 1.27 | .081, .890 | .079, .500 | .078, .320 | .073, .360 |
| | **SDE-GAN** | .190, 41.0 | .167, .000 | .243, .000 | .240, .000 | .236, .000 |
| | **FDM (ours)** | .006, 4.12 | .003, 1.63 | .002, .640 | .003, 1.18 | .004, 1.59 |
| GOLD | **SigKer** | .004, 1.95 | .004, 1.85 | .005, 2.13 | .006, 2.66 | .006, 2.53 |
| | **TruncSig** | .039, 9.49 | .039, 2.62 | .038, .510 | .034, .28 | .034, .130 |
| | **SDE-GAN** | .033, 12.0 | .072, 10.5 | .100, 6.20 | .131, 5.36 | .194, 8.21 |
| | **FDM (ours)** | .002, .900 | .002, .720 | .003, 1.05 | .004, 1.65 | .005, 2.01 |

We present additional qualitative results showing the pairwise joint dynamics generated by models trained on different datasets. Results for the metal price dataset are shown in Figure 3. Results for

Table 12: Standard deviations of average KS test scores and chance of rejecting the null hypothesis (%) at 5%-significance level on marginals of U.S. stock indices, trained on paths evenly sampled at 64 timestamps. "DOLLAR", "USA30", "USA500", "USATECH", and "USSC2000" stand for US Dollar Index, USA 30 Index, USA 500 Index, USA 100 Technical Index, and US Small Cap 2000, respectively.

| Dim | Model | t=6 | t=19 | t=32 | t=44 | t=57 |
|-----|-------|-----|------|------|------|------|
| DOLLAR | **SigKer** | .105, 30.9 | .168, 23.3 | .184, 20.0 | .186, 17.3 | .185, 14.7 |
|  | **TruncSig** | .081, 1.90 | .075, .365 | .067, .224 | .063, .134 | .058, .130 |
|  | **SDE-GAN** | .144, 15.4 | .218, 3.85 | .264, 1.07 | .256, .447 | .273, .346 |
|  | **FDM (ours)** | .032, 30.6 | .036, 33.9 | .037, 34.9 | .038, 35.3 | .038, 35.1 |
| USA30 | **SigKer** | .081, 31.7 | .106, 23.3 | .106, 6.00 | .110, 5.36 | .116, 6.09 |
|  | **TruncSig** | .050, 42.2 | .058, 46.6 | .056, 38.5 | .055, 29.3 | .054, 16.2 |
|  | **SDE-GAN** | .205, 29.2 | .328, 10.3 | .335, 17.6 | .266, .179 | .229, .045 |
|  | **FDM (ours)** | .008, 4.69 | .004, 2.76 | .003, 2.17 | .002, 1.74 | .001, 1.20 |
| USA500 | **SigKer** | .124, 16.2 | .191, 9.24 | .209, 6.53 | .215, 6.66 | .210, 7.04 |
|  | **TruncSig** | .064, 43.9 | .070, 43.9 | .070, 38.7 | .067, 33.4 | .061, 22.6 |
|  | **SDE-GAN** | .067, 4.03 | .246, 14.7 | .235, .045 | .252, .000 | .236, .000 |
|  | **FDM (ours)** | .005, 3.23 | .002, 1.17 | .001, .740 | .001, .682 | .001, .789 |
| USA1000 | **SigKer** | .030, 11.8 | .052, 7.61 | .059, 7.83 | .062, 7.47 | .057, 6.84 |
|  | **TruncSig** | .034, 34.1 | .037, 27.0 | .039, 17.6 | .042, 10.2 | .041, 4.54 |
|  | **SDE-GAN** | .056, .141 | .102, .000 | .090, .000 | .071, .000 | .056, .000 |
|  | **FDM (ours)** | .003, 1.75 | .001, .973 | .000, .554 | .001, .603 | .001, .598 |
| USA2000 | **SigKer** | .115, 27.8 | .164, 11.5 | .175, 5.85 | .183, 5.87 | .190, 5.90 |
|  | **TruncSig** | .062, 37.2 | .060, 35.4 | .058, 26.4 | .051, 13.7 | .046, 4.13 |
|  | **SDE-GAN** | .181, 32.4 | .296, 2.99 | .250, .045 | .203, .000 | .156, .000 |
|  | **FDM (ours)** | .007, 4.22 | .005, 2.90 | .003, 1.63 | .002, 1.51 | .002, 1.48 |

Table 13: Standard deviations of average KS test scores and the chance of rejecting the null hypothesis (%) at 5%-significance level on marginals for different currency pairs (EUR/USD and USD/JPY), trained on paths evenly sampled at 64 timestamps.

| Dim | Model | t=6 | t=19 | t=32 | t=44 | t=57 |
|-----|-------|-----|------|------|------|------|
| EUR/USD | **SigKer** | .134, 44.3 | .190, 39.8 | .215, 39.3 | .231, 39.1 | .239, 36.6 |
|  | **TruncSig** | .078, 1.97 | .071, .230 | .066, .148 | .061, .152 | .054, .089 |
|  | **SDE-GAN** | .218, 24.2 | .277, 9.44 | .309, 8.90 | .312, 5.23 | .287, .447 |
|  | **FDM (ours)** | .007, 4.91 | .002, 1.03 | .001, 0.627 | .002, 0.789 | .002, 0.852 |
| USD/JPY | **SigKer** | .066, 37.4 | .106, 38.3 | .126, 38.7 | .138, 39.8 | .138, 39.9 |
|  | **TruncSig** | .063, 19.7 | .062, 3.69 | .061, .841 | .056, .378 | .052, .173 |
|  | **SDE-GAN** | .051, 31.2 | .092, 25.3 | .147, 18.3 | .190, 16.4 | .237, 13.5 |
|  | **FDM (ours)** | .003, 1.51 | .001, 0.428 | .001, 0.410 | .001, 0.349 | .002, 0.522 |

the U.S. stock indices dataset are presented in Figures 4, 5, 6, 7, 8, 9, 10, 11, 12, and 13. Results for the exchange rates data are presented in Figure 14. Results for the energy price data are shown in Figures 15, 16, 17, 18, 19, and 20. Finally, results for the bonds data are presented in Figures 21, 22, and 23.

We present additional qualitative results comparing real and generated sample paths. Results for the exchange rates data are presented in Figures 24 and 25. Similarly, Figures 26 through 30 show the sample paths for five features from the stock indices dataset: "DOLLAR," "USA30," "USA500," "USATECH," and "USSC2000". Finally, Figures 31 and 32 depict the sample paths for silver and gold prices from the metal dataset. These plots demonstrate the ability of Neural SDEs to capture dynamics across diverse datasets.

Table 14: Standard deviations of average KS test scores and the chance of rejecting the null hypothesis (%) at 5%-significance level on marginals of energy prices, trained on paths evenly sampled at 64 timestamps. We reserve the latest 20% data as test dataset and measure how well the model predicts into future. "BRENT", "DIESEL", "GAS", and "LIGHT" stand for U.S. Brent Crude Oil, Gas oil, Natural Gas, and U.S. Light Crude Oil, respectively.

| Dim | Model | t=6 | t=19 | t=32 | t=44 | t=57 |
|---|---|---|---|---|---|---|
| BRENT | **SigKer** | .143, 41.6 | .217, 39.9 | .242, 42.4 | .238, 43.4 | .231, 44.1 |
| | **TruncSig** | .086, 5.70 | .087, 4.86 | .085, 4.40 | .086, 2.94 | .080, 1.94 |
| | **SDE-GAN** | .195, 6.65 | .133, .000 | .104, .000 | .068, .000 | .036, .000 |
| | **FDM (ours)** | .007, 2.77 | .006, 2.27 | .007, 4.08 | .007, 3.66 | .006, 2.87 |
| DIESEL | **SigKer** | .067, 31.0 | .112, 26.3 | .139, 28.5 | .158, 32.9 | .170, 35.9 |
| | **TruncSig** | .044, 21.3 | .044, 7.02 | .043, 2.22 | .042, .971 | .046, .404 |
| | **SDE-GAN** | .146, 28.7 | .277, 4.69 | .333, .303 | .302, .045 | .262, .000 |
| | **FDM (ours)** | .003, 1.25 | .002, 1.23 | .002, 1.29 | .002, 1.44 | .003, 1.47 |
| GAS | **SigKer** | .099, 27.5 | .157, 23.6 | .179, 30.5 | .182, 33.3 | .189, 36.6 |
| | **TruncSig** | .071, 32.8 | .069, 13.3 | .064, 4.65 | .063, .952 | .057, .311 |
| | **SDE-GAN** | .176, 23.2 | .290, .358 | .302, .268 | .251, .045 | .182, .000 |
| | **FDM (ours)** | .002, 0.99 | .004, 1.77 | .004, 2.12 | .005, 2.94 | .006, 3.67 |
| LIGHT | **SigKer** | .017, 18.6 | .039, 26.4 | .050, 32.1 | .048, 32.7 | .051, 34.4 |
| | **TruncSig** | .038, 16.5 | .040, 10.9 | .041, 6.92 | .042, 2.40 | .042, .728 |
| | **SDE-GAN** | .125, 30.4 | .349, 31.6 | .338, 21.2 | .268, 3.18 | .268, .179 |
| | **FDM (ours)** | .003, 1.38 | .003, 1.42 | .004, 2.05 | .006, 3.30 | .008, 4.97 |

Table 15: Standard deviations of average KS test scores and chance of rejecting the null hypothesis (%) at 5%-significance level on marginals of bonds, trained on paths evenly sampled at 64 timestamps. We reserve the most latest 20% data as test dataset and measure how well the model predicts into future. "BUND", "UKGILT", and "USTBOND" stand for Euro Bund, UK Long Gilt, and US T-BOND, respectively.

| Dim | Model | t=6 | t=19 | t=32 | t=44 | t=57 |
|---|---|---|---|---|---|---|
| BUND | **SigKer** | .126, 37.5 | .181, 32.6 | .190, 27.7 | .183, 24.7 | .179, 27.0 |
| | **TruncSig** | .077, 3.95 | .072, .523 | .068, .251 | .063, .130 | .054, .130 |
| | **SDE-GAN** | .090, 3.75 | .237, .045 | .282, 1.48 | .202, .000 | .218, .000 |
| | **FDM (ours)** | .008, 4.08 | .012, 4.78 | .008, 3.56 | .006, 2.38 | .007, 2.60 |
| UKGILT | **SigKer** | .094, 41.1 | .135, 40.7 | .146, 39.3 | .150, 38.5 | .152, 38.2 |
| | **TruncSig** | .050, 37.7 | .047, 16.5 | .043, 1.58 | .039, .336 | .037, .182 |
| | **SDE-GAN** | .232, 30.5 | .303, .394 | .212, .000 | .188, .000 | .156, .000 |
| | **FDM (ours)** | .003, 1.61 | .002, .606 | .001, 0.691 | .001, .611 | .002, .962 |
| USTBOND | **SigKer** | .096, 38.1 | .146, 39.4 | .159, 36.3 | .160, 36.2 | .158, 35.9 |
| | **TruncSig** | .077, 37.7 | .076, 30.1 | .071, 9.59 | .066, 1.18 | .057, .261 |
| | **SDE-GAN** | .162, 32.3 | .238, .045 | .207, .000 | .197, .000 | .188, .000 |
| | **FDM (ours)** | .006, 4.46 | .003, 1.40 | .001, 0.583 | .001, 0.560 | .001, .669 |

Table 16: Training time of different methods on forex data with different lengths in terms of hours. SDE-GAN hits the max wall times of 20 hours while the training progress is nearly 25%.

| Method | 64 Timestamps | 256 Timestamps | 1024 Timestamps |
|---|---|---|---|
| Signature Kernel | 0.66 | 7.80 | thread limit error |
| Truncated Signature | 0.31 | 1.34 | 5.61 |
| SDE-GAN | 0.64 | 4.21 | > 80 |
| FDM (ours) | **0.27** | **1.21** | **5.43** |

Table 17: Training time of Rough Bergomi model with different data dimensions in terms of hours.

| Method | 16 Dim | 32 Dim |
|---|---|---|
| SDE-GAN | 1.41 | 1.58 |
| FDM (ours) | **0.40** | **0.54** |
| Signature Kernel | 4.11 | 6.74 |
| Truncated Signature | 6.86 | GPU out of RAM |

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

 BRENT (U.S. Brent Crude Oil) and the vertical axis is LIGHT (U.S. Light Crude Oil).

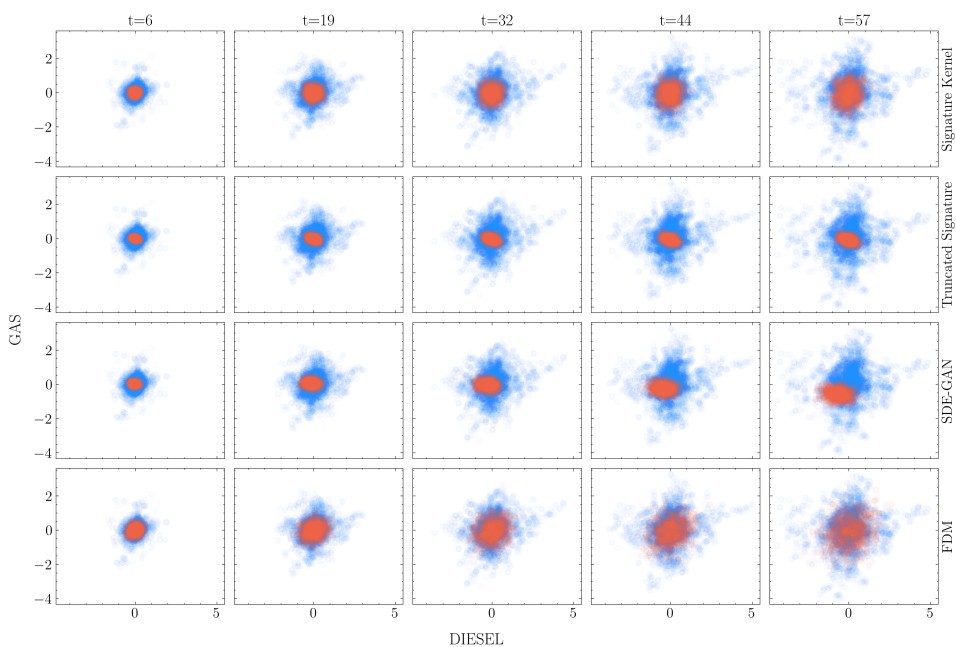

Figure 18: Blue points are real samples and orange points are generated by Neural SDEs. The dynamics of the joint distribution of DIESEL and GAS in energy data. Each row of plots corresponds to a method and each row corresponds to a timestamp. For each plot, the horizontal axis is DIESEL (Gas Oil) and the vertical axis is GAS (Natural Gas).

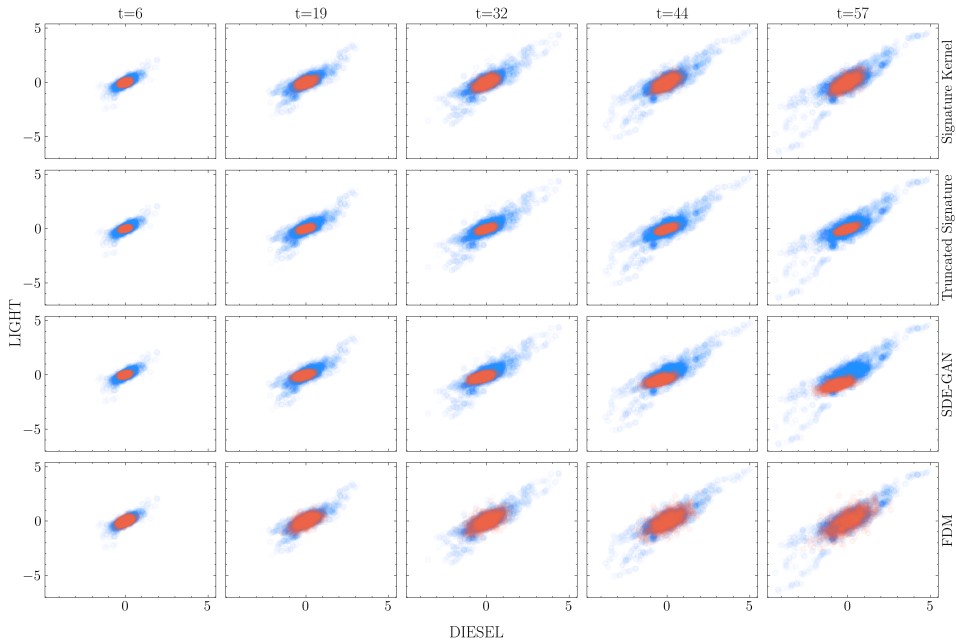

Figure 19: Blue points are real samples and orange points are generated by Neural SDEs. The dynamics of the joint distribution of DIESEL and LIGHT in energy data. Each row of plots corresponds to a method and each row corresponds to a timestamp. For each plot, the horizontal axis is DIESEL (Gas Oil) and the vertical axis is LIGHT (U.S. Light Crude Oil).

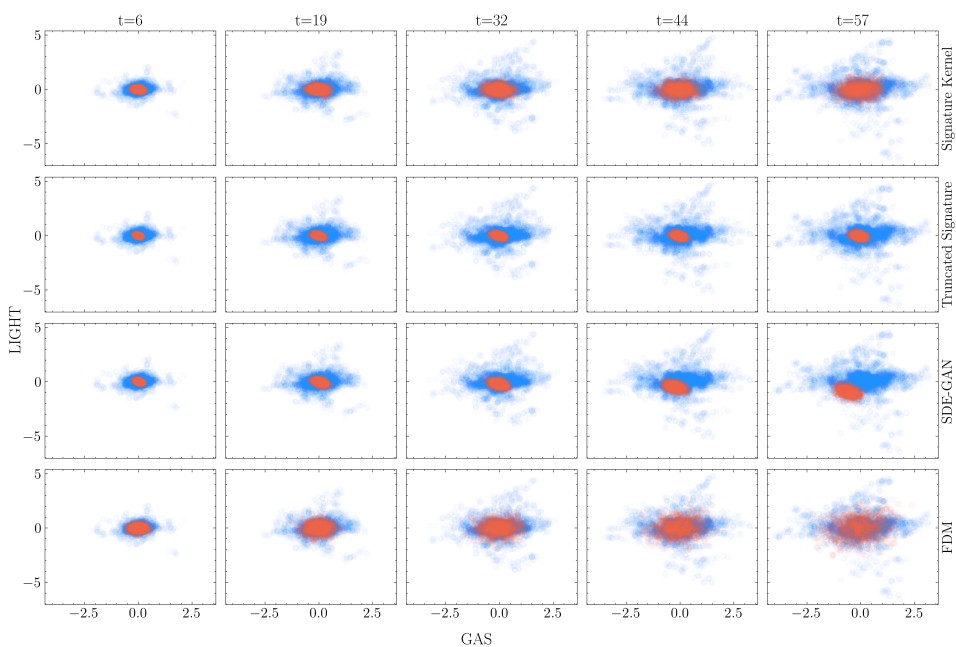

Figure 20: Blue points are real samples and orange points are generated by Neural SDEs. The dynamics of the joint distribution of GAS and LIGHT in energy data. Each row of plots corresponds to a method and each row corresponds to a timestamp. For each plot, the horizontal axis is GAS (Natural Gas) and the vertical axis is LIGHT (U.S. Light Crude Oil).

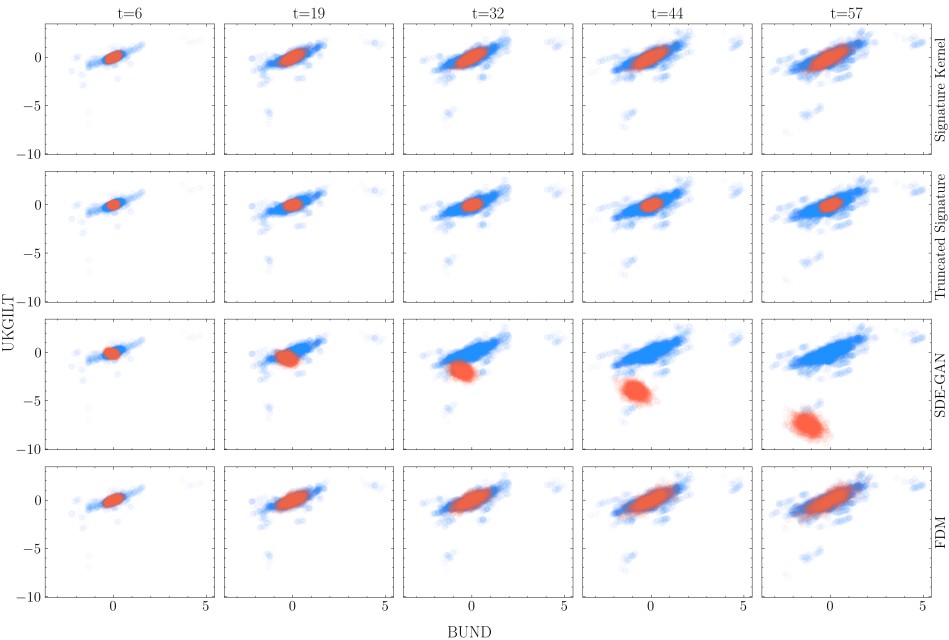

Figure 21: Blue points are real samples and orange points are generated by Neural SDEs. The dynamics of the joint distribution of BUND and UKGILT in bunds data. Each row of plots corresponds to a method and each row corresponds to a timestamp. For each plot, the horizontal axis is BUND (Euro Bund) and the vertical axis is UKGILT (UK Long Gilt).

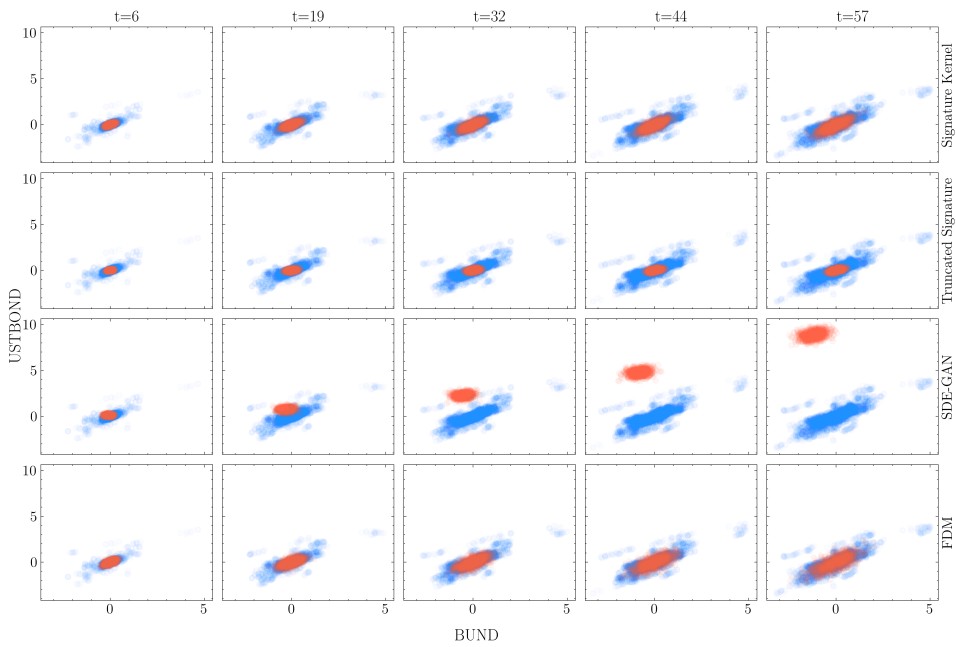

Figure 22: Blue points are real samples and orange points are generated by Neural SDEs. The dynamics of the joint distribution of BUND and USTBOND in bunds data. Each row of plots corresponds to a method and each row corresponds to a timestamp. For each plot, the horizontal axis is BUND (Euro Bund) and the vertical axis is USTBOND (US T-BOND).

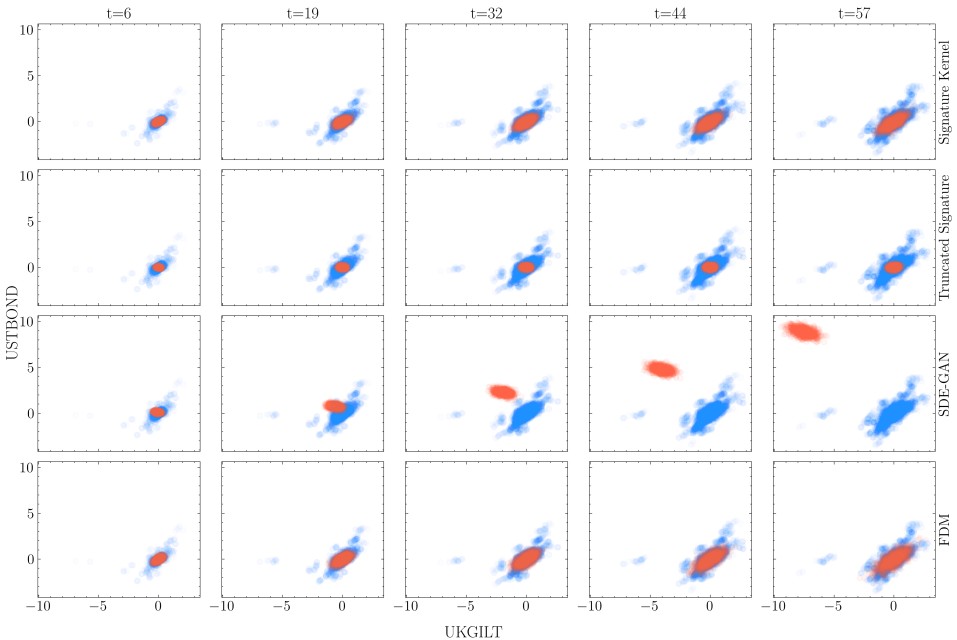

Figure 23: Blue points are real samples and orange points are generated by Neural SDEs. The dynamics of the joint distribution of UKGILT and USTBOND in bunds data. Each row of plots corresponds to a method and each row corresponds to a timestamp. For each plot, the horizontal axis is UKGILT (UK Long Gilt) and the vertical axis is USTBOND (US T-BOND).

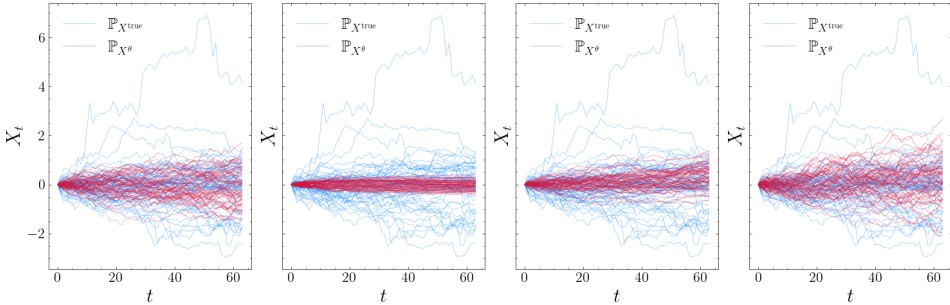

Figure 24: Sample paths for EUR/USD exchange rates from the exchange rate dataset. Blue lines represent real samples, while red lines represent those generated by Neural SDEs. From left to right, the plots correspond to signature kernels, truncated signature, SDE-GAN, and FDM, respectively. The horizontal axis represents time, and the vertical axis represents the EUR/USD exchange rate.

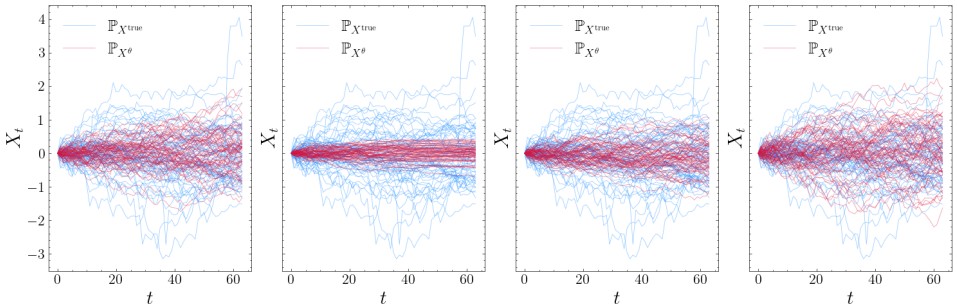

Figure 25: Sample paths for USD/JPY exchange rates from the exchange rate dataset. Blue lines represent real samples, while red lines represent those generated by Neural SDEs. From left to right, the plots correspond to signature kernels, truncated signature, SDE-GAN, and FDM, respectively. The horizontal axis represents time, and the vertical axis represents the USD/JPY exchange rate.

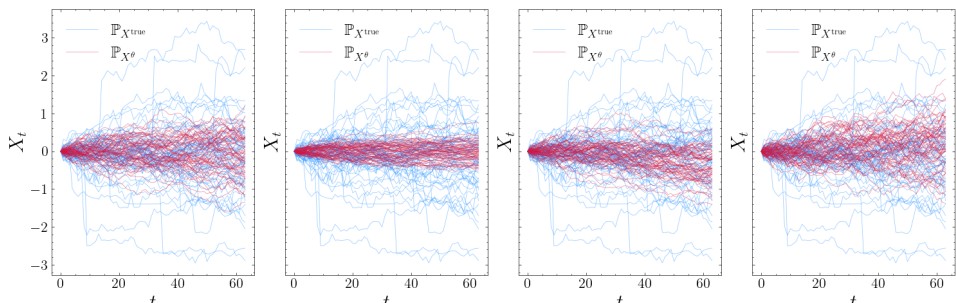

Figure 26: Sample paths for "DOLLAR" index from the stock indices dataset. Blue lines represent real samples, while red lines represent those generated by Neural SDEs. From left to right, the plots correspond to signature kernels, truncated signature, SDE-GAN, and FDM, respectively. The horizontal axis represents time, and the vertical axis represents the "DOLLAR" index value.

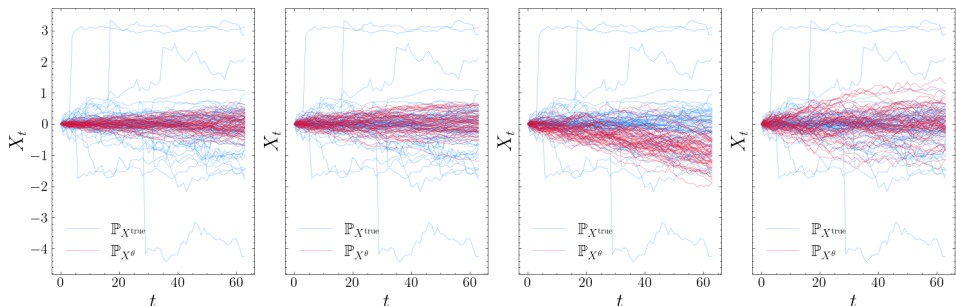

Figure 27: Sample paths for "USA30" index from the stock indices dataset. Blue lines represent real samples, while red lines represent those generated by Neural SDEs. From left to right, the plots correspond to signature kernels, truncated signature, SDE-GAN, and FDM, respectively. The horizontal axis represents time, and the vertical axis represents the "USA30" index value.

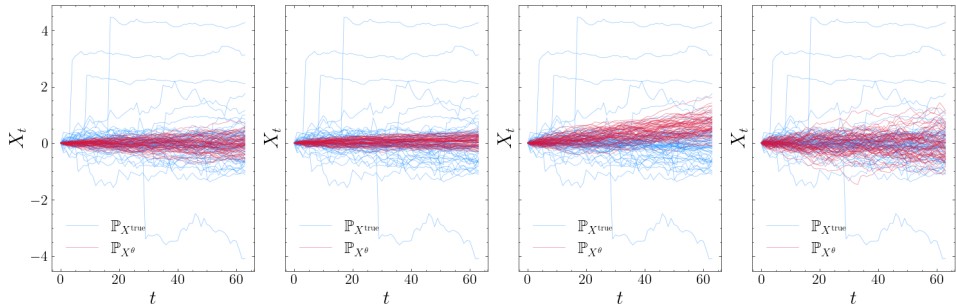

Figure 28: Sample paths for "USA500" index from the stock indices dataset. Blue lines represent real samples, while red lines represent those generated by Neural SDEs. From left to right, the plots correspond to signature kernels, truncated signature, SDE-GAN, and FDM, respectively. The horizontal axis represents time, and the vertical axis represents the "USA500" index value.

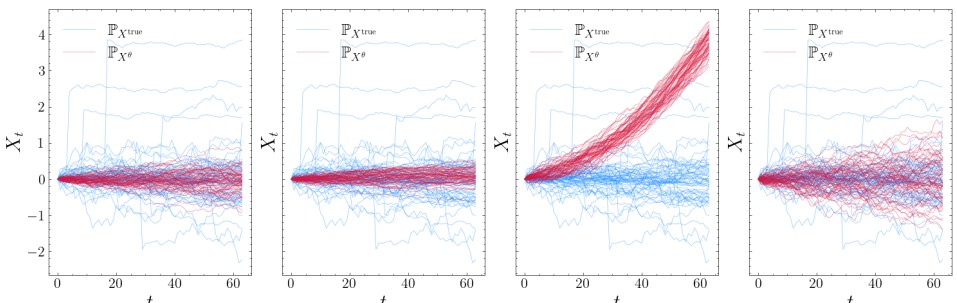

Figure 29: Sample paths for "USATECH" index from the stock indices dataset. Blue lines represent real samples, while red lines represent those generated by Neural SDEs. From left to right, the plots correspond to signature kernels, truncated signature, SDE-GAN, and FDM, respectively. The horizontal axis represents time, and the vertical axis represents the "USATECH" index value.

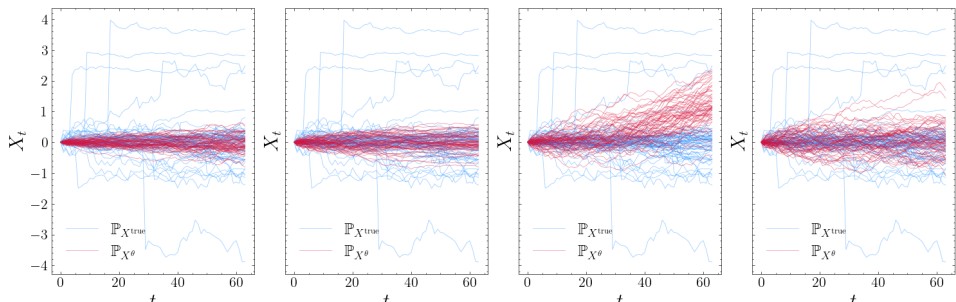

Figure 30: Sample paths for "USSC2000" index from the stock indices dataset. Blue lines represent real samples, while red lines represent those generated by Neural SDEs. From left to right, the plots correspond to signature kernels, truncated signature, SDE-GAN, and FDM, respectively. The horizontal axis represents time, and the vertical axis represents the "USSC2000" index value.

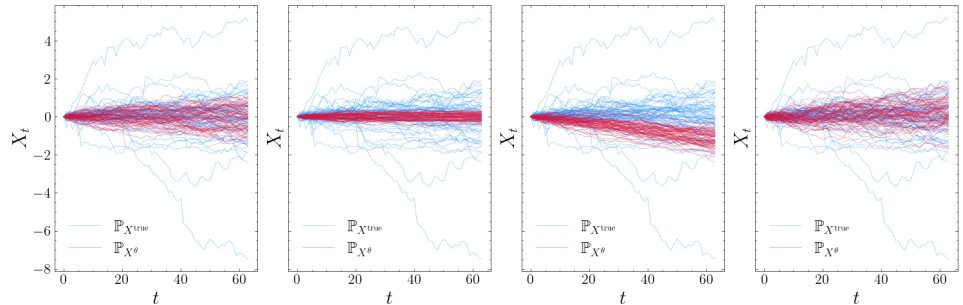

Figure 31: Sample paths for silver prices from the metal dataset. Blue lines represent real samples, while red lines represent those generated by Neural SDEs. From left to right, the plots correspond to signature kernels, truncated signature, SDE-GAN, and FDM, respectively. The horizontal axis represents time, and the vertical axis represents silver prices.

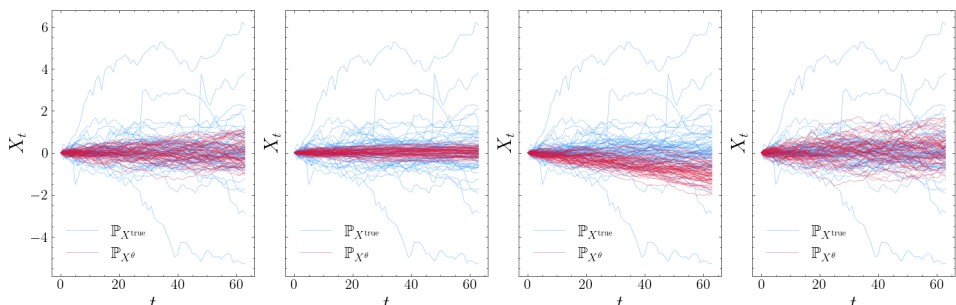

Figure 32: Sample paths for gold prices from the metal dataset. Blue lines represent real samples, while red lines represent those generated by Neural SDEs. From left to right, the plots correspond to signature kernels, truncated signature, SDE-GAN, and FDM, respectively. The horizontal axis represents time, and the vertical axis represents gold prices.