# OpenReview forum: "Efficient Training of Neural Stochastic Differential Equations by Matching Finite Dimensional Distributions"
_ICLR.cc/2025/Conference — ICLR 2025 Poster_

### Official Review · Reviewer_4f46 · 2024-10-28

**Soundness:** 1
**Presentation:** 2
**Contribution:** 2
**Rating:** 5
**Confidence:** 3

**Summary:**

This paper presents a novel approach to training neural stochastic differential equations by introducing Finite Dimensional Matching, a new scoring rule designed for continuous Markov processes. The proposed method leverages the Markov property of stochastic processes to reduce the computational complexity of training neural SDEs, with the goal of enhancing both efficiency and generative quality. Theoretical contributions establish that the new scoring rule provides a strictly proper method for comparing two-time joint distributions. Experimental results demonstrate improved performance in training efficiency and generative quality when evaluated against competing methods.

**Strengths:**

The paper tackles a crucial challenge in training neural SDEs by introducing a new scoring rule that optimizes efficiency for continuous Markov processes. The proposed FDM method is backed by mathematical proofs, providing a strictly proper scoring rule that extends from finite-dimensional distributions to continuous Markov processes. This theoretical contribution is valuable to the literature on neural SDE training methods. Experiments show that FDM offers computational efficiency gains, reducing training complexity from quadratic to linear in the number of discretization steps. The approach outperforms prior methods in computational efficiency, as shown in multiple experimental benchmarks.

**Weaknesses:**

The paper’s main theorem relies on strong assumptions regarding the Markovian properties and continuity of the processes involved. These assumptions may limit the applicability of the FDM algorithm in more complex, non-Markovian stochastic processes or those with jumps, which are common in real-world scenarios. Explicitly discussing these limitations and potential ways to relax these assumptions would make the contributions more transparent.

Also, the writing template is for ICLR 2024, not ICLR 2025.

**Questions:**

Please see weaknesses.

---

> ### Author Response · Authors · 2024-11-12
>
> Hi,
>
> Thank you for the review, and we're happy to address the raised concerns.
>
> 1. We're going to fix the template in the forthcoming revision.
>
> 2. *Continuity and Markov Assumption*
>
> As mentioned in the response to reviewer SBGZ ([link](https://openreview.net/forum?id=d4qMoUSMLT&noteId=zKB258o4r6)), the proof can be directly extended to cadlag Markov processes defined on an open interval $[0, T)$. The only modification needed in the proof is to construct the sequences to converge from the right in the last paragraph of the proof of Theorem 5. If you and the other reviewers deem it necessary, we are happy to extend the proof to the cadlag case. This allows our theory to cover jump processes.
>
> Regarding the Markov assumption, we believe this can be partially addressed by augmenting the paths with more information or leveraging a hidden Markov model. However, these are beyond the scope of neural SDEs and may warrant a separate paper. If you feel it is necessary, we can add additional discussion on the relaxation of these assumptions.
>
> We thank you again for your insights, and please let us know if there is anything unclear; we're more than happy to clarify!

---

> > ### Comment · Reviewer_DrUo · 2024-11-14
> >
> > In my opinion, the "strong assumptions regarding the Markovian properties and continuity of the processes involved" mentioned by reviewer 4f46 are relatively acceptable for many applications - for instance stochastic modelling in biology and healthcare, with which I am fairly familiar. While I am not familiar with non-Markovian processes, I have never encountered them in my research. Do you have any examples in mind ? I am genuinely interested in this question.
> >
> > I do agree however that including jumps is of high interest. The authors could maybe at least empirically verify that their scoring rules are effective for training neural jump SDE (see https://arxiv.org/pdf/1905.10403) on a simple example. The code of this paper is available (https://github.com/000Justin000/torchdiffeq/tree/jj585) and seems to be well-built. However, I am unsure whether such an extension can be reasonably implemented during the short reviewing period.

---

### Official Review · Reviewer_DrUo · 2024-10-29

**Soundness:** 4
**Presentation:** 2
**Contribution:** 3
**Rating:** 8
**Confidence:** 4

**Summary:**

The authors propose a scoring rule for continuous time stochastic processes that is directly derived from a scoring rule on a generic space. They show that this rule is proper (i.e. injective from the space of paths to the space of laws), which is a non-trivial contribution. Experiments show that this method outperforms existing concurrent methods based on signature kernels and SDE-GANs. I stress that the experiments are carried out on a vast array of datasets.

**Strengths:**

* The paper is well written and structured, making it easy to read and to follow. The introduction is sound and features a thorough literature review.

* The main technical contribution is Theorem 2, which allows to convert any scoring rule on a generic space. I believe this contribution to be non trivial and novel, although simple to prove.

* The experiments are carried out on a vast array of time series datasets and show overall superior performance of the proposed approach. The authors compare themselves to all relevant baselines to the best of my knowledge. Large generative models such as diffusion models are not included ; however, I do believe that they do not belong to the same class of models and do not require a comparable computational budget.

**Weaknesses:**

* While the contribution made in Theorem 2 is elegant and novel, one could object that it is slightly insufficient --- this is not my point of view, but another reviewer might disagree and I am willing to discuss this point. In order to strengthen their theoretical part, I encourage the authors to consider for instance the sample complexity of their kernel i.e. how fast does the empirical divergence they define through a kernel on $\mathcal{E}$ converge through the expected divergence ? Do sample complexities in $\mathcal{E}$ carry over to the space of paths ? See Gretton (2012) for an example.

* A blind spot of the paper is in my opinion the choice of the kernel. The authors do not seem to consider other kernels for $\mathbb{R}^d$ valued processes, which could yield an interesting extension. This might especially be interesting since some kernels are sometimes used because of their specific properties (invariances, ...). I would suggest that the authors consider this point, at least in the Appendix.

* This approach nicely extends to kernels defined on any space - this could be graphs, images, etc. and could allow to generate time series in these spaces. This would provide, in my sens, a extremely valuable extension to the paper.

* The experimental section is hard to read and a tad unstructured. I encourage the authors to use less tables, add more comments and broaden the analysis of their experiments. Also, a notable restriction of this part is that experiments are only carried out on $2$ dimensional time series. I strongly encourage the authors to extend their experiments to high dimensional datasets. Also, there are no confidence intervals in the tables.

* Regarding this last point, I believe that a valuable extension could be to consider random feature approximations to the kernels for high dimensional generation, which is still a major hurdle in the field. Similarly, the authors could consider sliced kernels on $\mathcal{E}$ when the dimension is high.

* Concerning experiments, an interesting task to consider could be the augmentation of a time-series dataset, and the analysis of the gain in performance for any model trained on this dataset.

* Concerning experiments, I believe that it would be highly valuable to extend applications beyond finance. Generating time series is a major hurdle in many domains with great social impact such as neuroscience, healthcare, biology, climatology, economics ...

* A valuable extension of this work would be the investigate the use of the devised score for other purposes than training generative models, such as two-sample tests for instance.

**Questions:**

* Please include confidence intervals in your tables: variability of your results is a very important aspect.

* Could you please include vizualizations of the full generated time series, rather than only plotting the time marginals ?

* Could you add experiments on at least one high dimensional and one non euclidian real-world dataset ?

I would considered increasing my score if a significant number of concerns on the experiments are addressed. Similarly, I could consider lowering my score during the discussion phase if other reviewers relativise the strength of the theoretical contribution --- which again seems sufficient to me.

---

> ### Author Response · Authors · 2024-11-14
>
> Hi,
>
> Thank you for the detailed and constructive review. We're happy to address the raised concerns.
>
> **1. Experiments are only carried out on 2-dimensional time series**
>
> We apologize for the confusion, but this is not the case. For each dataset, we model all the features jointly, i.e., the dimension is $2$ for the metal price and exchange rate dataset, $5$ for the stock indices dataset, $4$ for the energy price dataset, $3$ for the bonds dataset, and $16$ and $32$ for the Rough Bergomi Model data. We present the two-dimensional joint marginals only because plotting higher-dimensional distributions on paper is challenging.
>
> **2. Visualizations of the full generated time series**
>
> Thank you for pointing this out. We'll add them in the forthcoming revision.
>
> **3. Non-Euclidean real-world dataset**
>
> We're not familiar with this type of dataset that is suitable to be modeled as an SDE. We would appreciate any recommendations you might have.
>
> **4. Choice of kernels**
>
> We can include a study on the kernel choice in the appendix in the forthcoming revision.
>
> **5. Augmentation of a time-series dataset**
>
> We're not sure what this means; could you please elaborate? Do you mean using the generative method for data augmentation and applying it to a downstream task?
>
> **6. Random feature approximations to the kernels for high-dimensional generation**
>
> Could you please elaborate on this as well? Do you mean computing the kernels with random feature approximation? It seems like this is just another way to compute/approximate the kernel.
>
> **7. Sample complexity**
>
> We agree that this is an interesting aspect and will look into it. We'll try to integrate sample complexity analysis in the final revision by the end of the discussion period, but we cannot guarantee.
>
> **8. Confidence intervals**
>
> Repeating some models on higher-dimensional datasets or datasets with longer sequences can be computationally expensive, so we omitted the confidence intervals for consistency. The previous state-of-the-art [Issa et al., 2023](https://arxiv.org/pdf/2305.16274) in this area also didn't report error bars, so we believed this was acceptable. That being said, we're happy to include confidence intervals for some smaller datasets in the appendix in the forthcoming revision.
>
> We thank you again for your insights, and please let us know if there is anything unclear; we're more than happy to clarify!

---

> > ### Comment · Reviewer_DrUo · 2024-11-14
> >
> > Thank you for your answers on points **1,2 and 4**.
> >
> > **Concerning point 3**, you can apply your method to any space on which you have a suitable kernel if I am not mistaken. This paper develops neural SDEs for graphs for instance https://arxiv.org/pdf/2308.12316 and could give you some inspiration. You could also think of shapes evolving through time, a topic of interest in healthcare (think about organs and cells deforming over time). Finally, you could also consider measures evolving over time - see for instance https://proceedings.mlr.press/v151/bunne22a/bunne22a.pdf.
> >
> > **Concerning point 5**, yes this is exactly what I mean. I believe that this would allow you to show that generating time series for training a downstream model with your method has real benefits (which is something that I don't believe to be straightforward, see for instance https://arxiv.org/abs/2402.07712v1).
> >
> > **Concerning point 6**: even if your method simplifies the optimization of the neural SDE through a simpler kernel (compared to the signature kernel which can be technical to compute with high precision, or the GAN-like training of Kidger et al 2021), you still have to evaluate this very kernel. To simply such a computation, a long lasting line of research going back to Rahimi (2007) has resorted to randomised approximations. I believe that this framework could allow you to train neural SDEs for generating high dimensional time series.
> >
> > **Concerning point 7**: my intuition would be that the sample complexity is similar. But then you need to approximate the expectation in definition 1, so you might have to deal with a concentration bound on this stochastic approximation. Hence intuitively I believe that you should end up with the classical sample complexity of kernel mean embeddings + a term that depends on the number of times you draw times to compute your scoring function. Could you maybe provide an empirical assessment of this ? A plot of your metric vs the number of samples used and a plot of your metric vs the number of times drawn would suffice to convince me.
> >
> > **Concerning point 8**: I sincerely believe that this is not an acceptable answer. Reporting confidence intervals is a common and well-established scientific practice. NeurIPS, for instance, requires it in the paper checklist: https://neurips.cc/Conferences/2022/PaperInformation/PaperChecklist. Providing confidence intervals on even smaller datasets is a bare minimum.

---

> > > ### Author Response · Authors · 2024-11-14
> > >
> > > Thank you again for the insightful review.
> > >
> > > **Concerning point 3**, thank you for the reference. We'll try to include experiments on non-Euclidean datasets in the final revision.
> > >
> > > **Concerning points 5 and 6**, I agree these are interesting directions. However, they seem to be beyond the scope of neural SDEs and may deserve a separate paper.
> > >
> > > **Concerning point 7**, thank you for the insight. The sample complexity bound is currently on our to-do list. We hope to integrate it into the final revision.
> > >
> > > **Concerning point 8**, we appreciate this advice and agree to add error bars at least for some smaller datasets. However, we do not have access to our cluster in the next few days, so the error bars will be included in a later revision toward the end of the discussion period.
> > >
> > > Again, we appreciate your time and insights.

---

> > > > ### Comment · Reviewer_DrUo · 2024-11-15
> > > >
> > > > Many thanks for your answers. I look forward to reading your revision. Let me know if you want to discuss any other point of your paper.

---

### Official Review · Reviewer_SGBZ · 2024-11-01

**Soundness:** 4
**Presentation:** 2
**Contribution:** 3
**Rating:** 8
**Confidence:** 4

**Summary:**

This paper introduces a very simple yet elegant way of comparing the similarity of the laws of two homogeneous Markov processes. The idea follows from the fact that the distribution of the process is completely determined by the transition kernel. Hence the ability to compare the transition distribution should give enable a comparison between the laws.

The authors then use this finding as a scoring rule to learn to simulate from a neural SDE, given the observations from a ground truth process. Then, they test their scoring rule and their FDD matching procedure in numerical experiments, showing that their method outperforms existing law matching techniques.

**Strengths:**

The method is very simple and elegant and the implication of this result, **if true**, could be impactful to the neural SDE and generative modeling community.

**Update: I was wrong about the validity of the proof. The result is sound. **

**Weaknesses:**

I think there is a significant issue in the proof that could invalidate your main theorem 2. In the middle part of page 14, you have the integral over $t_1,t_2$ of the $S(P_{t_1,t_2}, P_{t_1,t_2}')$ scoring rules being equal, but then you conclude that $$S(P_{t_1,t_2},P_{t_1,t_2}') = S(P_{t_1,t_2},P_{t_1,t_2})$$ a.e.. This is not true or I am missing something?

**Update: I was wrong about the validity of the proof. The result is sound. **

My intuition is that this is not an easy fix: You want to match the distribution over all $t_1,t_2$, so, in some sense, you want the expectation equality to hold for all test measures $\nu$ instead of just one particular $\nu$. I would be very happy to raise the score and rewrite my review if I am wrong. However, it does seem that proving a result like yours is possible, given that the generator or the resolvent will determine the law of the Markov process (see Either and Kurtz).

I appreciate the rigor in defining the math notations and results. But the writing and explanations in the paper need improvements. For example, what do you mean by "Update the model parameters $\theta$ through backpropogation to maximize $\hat S$" (inside the algorithm)? Also, there is no explanation of what the data is. What are the "Average KS test scores"? Are you repeating the experiment across multiple batches and producing the percentage of rejection "chance of rejecting the null hypothesis (%) at 5%-significance level on marginal"?

Other minor issues include:
1. I think you need $\mathcal{E}$ to be Polish.
2. The radial basis function (RBF) kernel is not defined.
3. The $\pi$ notation seems a little distracting, why not just use $(x_t,x_s)$ for $\pi_{t,s}(x)$.
4. It would be nice to emphasize that the Markov processes you define are homogeneous Markov processes.

**Questions:**

The main concern I had was in regard to the proof. Please help me to understand or fix the issue.

If this and the writing issues I raised above can be addressed, I will happily give you at least a 6. However, since I feel that your main result is wrong, I have to give a low score for now.

**Update: I was wrong about the validity of the proof. The result is sound. **

If this (or a variant of the) scoring rule for Markov processes is indeed correct, I think the authors could improve the paper by exploring the sensitivity properties of this scoring rule; for example, what kernel to use and how the score behaves when P and Q are close? Is there a simple formula to compute the gradient?

---

> ### Author Response · Authors · 2024-11-12
>
> Hi,
>
> Thank you for the detailed review and we're happy to address the raise concerns.
>
> 1. For the proof, it is important that we choose $\nu$ to be an equivalent measure to the Lebesgue measure $\mu$ on $[0, T] \times [0, T]$. So there exists a $\mu$-a.e. positive function $\lambda(t_1, t_2)$ such that $d\nu = \lambda d\mu$. (in case you need a proof, see https://math.stackexchange.com/questions/1393425/equivalent-finite-measures-if-and-only-if-strictly-positive-radon-nikodym-deriv)
>
> Then
> $$E_{(t_1, t_2) \sim \nu} S(P_{\pi_{t_1, t_2} (X)}, P_{\pi_{t_1, t_2} (Y)}) = E_{(t_1, t_2) \sim \nu} S(P_{\pi_{t_1, t_2} (Y)}, P_{\pi_{t_1, t_2} (Y)})$$
> suggests
> $$E_{(t_1, t_2) \sim \mu} \lambda(t_1, t_2) S(P_{\pi_{t_1, t_2} (X)}, P_{\pi_{t_1, t_2} (Y)}) = E_{(t_1, t_2) \sim \mu} \lambda(t_1, t_2) S(P_{\pi_{t_1, t_2} (Y)}, P_{\pi_{t_1, t_2} (Y)}),$$
> which further implies
> $$E_{(t_1, t_2) \sim \mu} \lambda(t_1, t_2) [ S(P_{\pi_{t_1, t_2} (Y)}, P_{\pi_{t_1, t_2} (Y)}) - S(P_{\pi_{t_1, t_2} (X)}, P_{\pi_{t_1, t_2} (Y)})]  = 0.$$
>
> Recall that by definition of the proper scoring rule $S(P_{\pi_{t_1, t_2} (Y)}, P_{\pi_{t_1, t_2} (Y)}) \geq S(P_{\pi_{t_1, t_2} (X)}, P_{\pi_{t_1, t_2} (Y)})$ and $\lambda(t_1, t_2) > 0$ $\mu$-a.e. due to the equivalence of $\nu$ and $\mu$. This makes the integrand to be non-negative $\mu$-a.e., forcing $\lambda(t_1, t_2) [ S(P_{\pi_{t_1, t_2} (Y)}, P_{\pi_{t_1, t_2} (Y)}) - S(P_{\pi_{t_1, t_2} (X)}, P_{\pi_{t_1, t_2} (Y)})]  = 0$ $\mu$-a.e..
>
> Again, due to the fact that $\lambda(t_1, t_2) > 0$ $\mu$-a.e., $S(P_{\pi_{t_1, t_2} (Y)}, P_{\pi_{t_1, t_2} (Y)}) - S(P_{\pi_{t_1, t_2} (X)}, P_{\pi_{t_1, t_2} (Y)}) = 0$ $\mu$-a.e..
>
> I think two important points here are the equivalence of between $\mu$ and $\nu$ and that $s$ is a strictly proper scoring rule.
>
> 2. I'm checking where we need $\mathcal{E}$ to be Polish. The disintegration theorem we used (Theorem 8.5 of Kallenberg (2021)) only requires the value space to be Borel. We'd sincerely appreciate it you can help us to point it out.
>
> 3. For the other minor issues, we can fix them in the upcoming revision.
>
> We thank you again for the insights and please let us know if you feel there is anything unclear; we're more than happy to clarify!

---

> > ### Comment · Reviewer_SGBZ · 2024-11-12
> > **Response to the clarification**
> >
> > I see, I forgot about the property of the scoring rule.
> >
> > By the way, in responding to the concerns of the other reviewer about the jumps, I think this should work for (at least a large subclass of) Feller processes. Because essentially the law of the Feller process will be determined by the generator. This essentially means that you only need to match the joint distribution over an infinitesimal interval. So, the a.e. Lebesgue would suffice.
> >
> > In this case, I will raise my score to 5 for now. If you can clarify your paper as suggested by me and other reviews, I will further improve the score to 6.
> >
> > As the reviewer DrUo helpfully pointed out, this would be a better submission if you could say something about the qualities/properties of the scoring rule, especially sensitivity properties,  i.e. how perturbations in $\mu$ and $\sigma$ would translate to $S$.

---

> > > ### Author Response · Authors · 2024-11-12
> > >
> > > Thank you for the quick response!
> > >
> > > Indeed, the proof can be directly extended to cadlag Markov processes defined on an open interval $[0, T)$. The only modification needed in the proof is to construct the sequences to converge from the right in the last paragraph of the proof of Theorem 5. As the focus of the paper is on neural SDEs, we did not include the proof for the cadlag case. That being said, if you and the other reviewers feel it is necessary, we are happy to extend the proof to cover the cadlag case.

---

> > > ### Author Response · Authors · 2024-11-12
> > >
> > > For the sensitive properties, could you recommend any literature? We'd deeply appreciate it.

---

> ### Comment · Reviewer_SGBZ · 2024-11-12
> **Polish and sensitivity**
>
> I misinterpreted your $X,Y$ taking value in $\mathcal E$. I thought you were saying that $\omega\rightarrow X(\omega,\cdot )$ is in $\mathcal E$. In any case, you do need the processes to be separable (continuity would suffice), i.e. the FDD is determined by a dense subset of $[0,T)$, to argue for the last part of Theorem 5. This is not explicitly stated in the assumption. I would suggest you just go with $X\in C_{\mathcal E}[0,T]$ or $D_{\mathcal E}[0,T]$ where $\mathcal E$ is a Euclidian space.
>
> For the sensitivity, I am wondering if the scoring rule is smooth (and how smooth) in a change in the parameter of the neural SDE. This will affect the convergence rates of your algorithm. I think this is interesting especially for the volatility part, as we know that the KL is infinity if you perturb the volatility (the two laws are not absolutely continuous).

---

> > ### Author Response · Authors · 2024-11-12
> >
> > Got it. I agree wo do need the separability for the last paragraph in the proof of Theorem 5. I'll add the Polish assumption for sure.
> >
> > The reviewer DrUo mentioned  $\mathcal{E}$ could be a space of graphs, so would $\mathcal{E}$ being Euclidian a bit too restrictive?
> >
> > Thank you again for the insightful and timely comments.

---

> > > ### Comment · Reviewer_DrUo · 2024-11-14
> > >
> > > Thanks to reviewer SGBZ and the authors for this discussion. I have myself also read carefully read the proof, and do no see any issues. This strengthens my opinion that the contributions of the paper are strong and of high interest.

---

> > > ### Comment · Reviewer_SGBZ · 2024-11-14
> > >
> > > It is your decision, but then you need to argue things through separability. I thought the focus of this paper should be providing a divergence for neural SDE applications. How would jump processes of graphs fit into the neural SDE context?

---

### Official Review · Reviewer_GLJM · 2024-11-08

**Soundness:** 2
**Presentation:** 1
**Contribution:** 2
**Rating:** 3
**Confidence:** 3

**Summary:**

This paper proposes a new method called Finite Dimensional Matching (FDM) for training Neural SDEs by identifying a class of strictly proper scoring rules for comparing continuous Markov processes. Using this scoring rule, they claim to reduce the training complexity from quadratic in discretization timesteps to linear.

**Strengths:**

This paper addresses an important problem: the high computational complexity (quadratic in time steps) associated with training Neural SDEs using scoring rules. The authors propose a reduced complexity method, aiming for linear complexity to enhance performance. They also provide a theoretically grounded approach in designing a new scoring rule. However, despite this strong motivation, the results are somewhat unconvincing due to certain weaknesses noted below.

**Weaknesses:**

This paper has several notable weaknesses. While the authors propose a reduced complexity approach for training Neural SDEs, they do not adequately explain key concepts, making the paper difficult to follow, especially for readers without a strong background in this area. For example, despite experience with SDEs in score-based generative models and deep learning theory, I found the explanations lacking in detail and context and some results are not convincing.

The authors do not provide sufficient preliminary material on scoring rules or background on how these rules are used to measure divergence between two Markov processes. A concrete example of the scoring rule $s(P, z)$ with an RBF kernel, presented early on, would have improved clarity.

Additionally, the complexity reduction claim is unconvincing. For example, at the beginning, the authors claim they reduce the complexity to linear using $D$ to denote the time steps, however, this notation $D$ is never used again in the rest of the paper. Instead, in Section 4.2 Algorithm, they compare two stochastic processes and use $B$ to denote the total number of time steps. However, the nested summations in the top equation on page 5 suggest quadratic rather than linear complexity, i.e., $B^2$.

Finally, the paper uses the **ICLR 2024 format rather than the ICLR 2025** format.

**Questions:**

In Theorem 2, does the result hold for any scoring rule $s$, or does $s$ also need to be strictly proper? Could you clarify if there are specific conditions on $s$ required for Theorem 2 to apply?

---

> ### Author Response · Authors · 2024-11-12
>
> Hi,
>
> We deeply appreciate the detailed reviews and are happy to address the concerns.
>
> *1. In Theorem 2, does the result hold for any scoring rule $s$, or does $s$ also need to be strictly proper?*
> Yes. $s$ has to be strictly proper. This is stated in the first paragraph of section 4.1: "*Let $s$ be any strictly proper scoring rule defined on ...*". We agree this is probably not sufficiently clear and will move this condition to the theorem statement in the forthcoming revision.
>
> *2. The complexity reduction claim*
> We apologize for this confusion. $D$ refers to the discretization step in the numerical integration of the SDE, i.e., for the SDE
> $$
> dZ_t = \mu^{\theta}(t, Z_t)  dt + \sigma^{\theta}(t, Z_t)  dW_t,
> $$
> we evaluate the integral
> $$
> Z_T = Z_0 + \int_{0}^{T} \mu^{\theta}(t, Z_t)  dt + \int_{0}^{T} \sigma^{\theta}(t, Z_t)  dW_t
> $$
> by numerical integration with $D$ discretization steps using the Euler-Maruyama method:
> $$
> Z_{t_{k+1}} = Z_{t_k} + \mu^{\theta}(t_k, Z_{t_k}) \Delta t + \sigma^{\theta}(t_k, Z_{t_k}) \Delta W_k
> $$
> where $\Delta t = T/D$, $t_{k+1}=t_k + \Delta t$, and $\Delta W_k$ is the Wiener increment over the interval $[t_k, t_{k+1}]$.
>
> $B$ in Algorithm 1 refers to the batch size of the SGD optimizer and is different from $D$.
>
> The $O(D^2)$ complexity comes from the previous state-of-the-art Neural SDE training method proposed in [Issa et al., 2023](https://arxiv.org/pdf/2305.16274), which requires solving a PDE
> $$
> f(s, t) = 1 + \int_{0}^{s} \int_{0}^{t} f(u, v) \langle dx_u, dy_v \rangle_1  dv du
> $$
> (see (2) in the linked paper) and backpropagate the gradients through the PDE solver. The double integral is typically numerically approximated using a rectangular rule with $D$ discretization steps:
> $$
> \int_{0}^{T} \int_{0}^{T} f(u, v) \langle dx_u, dy_v \rangle_1 dv du \approx \sum_{i=1}^{D} \sum_{j=1}^{D} f(u_i, v_j) \langle dx_{u_i}, dy_{v_j} \rangle \Delta u \Delta v,
> $$
> where $\Delta u = T/D$, $\Delta v = T/D$, and $u_i = i \Delta u$, $v_j = j \Delta v$ for $i, j = 1, \dots, D$. The double sum requires $O(D^2)$. Also, the method of [Issa et al., 2023](https://arxiv.org/pdf/2305.16274) does not have a better complexity on $B$ as their objective also involves a double sum over $B$ (see (4) in the linked paper, our $B$ is their $m$, the double integral occurs in their $k_{sig}$).
>
> So overall, our method reduces the complexity from $O(D^2)$ to $O(D)$ (or, from $O(D^2 B^2)$ to $O(D B^2)$ if you prefer to also include $B$ ) as we don't need to solve the PDE with the double integral.
>
> *3. Not sufficient preliminary material on scoring rules or background on how these rules are used to measure divergence between two Markov processes.*
> Thank you for pointing out the lack of clarity. We're happy to add more explanation to the background of scoring rules in the forthcoming revision. We'll move the RBF kernel example earlier.
>
> We thank you again for the insights and please let us know if you feel there is anything unclear; we're more than happy to clarify!

---

> > ### Comment · Reviewer_GLJM · 2024-11-25
> >
> > Thank you for your response. However, I did not notice any updates in the revised version. Based on the current write-up, I am not comfortable recommending acceptance for this paper. While the paper addresses an interesting problem, the current presentation is not ready for publication. I suggest that the authors undertake at least one round of revisions to significantly improve the clarity and quality of the presentation.

---

### Comment · Reviewer_DrUo · 2024-12-02
**Additional Comment**

As the end of the discussion period is approaching, I would like to stress once again to fellow reviewers and chairs that this paper substantiality contributes IMO to the field of neural SDEs and time series. The theoretical result presented in this paper hugely simplifies the current go-to pipeline, which often relies on the signature kernel whose computation is relatively expensive and does not scale to high dimensions. Applications to generative modelling display strong improvements over the current SOTA.

The authors have addressed various points during the discussion period and have improved their paper.

The potential applications, while not fully investigated by the authors, are considerable and include generative models, two-sample tests, kernel methods, transfer learning and the extension of all the previous to sequential data that lives in any space that can be endowed with a characteristic kernel.

I strongly believe that this paper should be accepted.

---

### Meta-Review · Area_Chair_JVYJ · 2024-12-21

**Metareview:**

In this work, the authors show that proper scoring rules on distributions can be extended to proper scoring rules on processes through their finite dimensional distributions. Using this result, a method called Finite Dimensional Matching (FDM) is proposed to bypass pain points with fitting stochastic processes to data. In a large number of numerical experiments, the authors demonstrate the advantages of this approach over competing methods. Reviewer opinions are mixed: two are positive, one mixed, and one negative. The reviewers stating positive opinions have stood firm on their stance that the paper should be accepted, while the more negative reviewers have failed to adequately engage in the discussion period. This is unfortunate, as the discussion has been considerable!

My own stance agrees with the positive feedback: the results are simple in retrospect, but profound in development and application. This is a strong contribution to the literature, and I believe it is worthwhile to the community even in its current state. **However**, in addition to reviewer feedback, I object to several aspects of the presentation: text in figures is unacceptably small (should be close to same size as surrounding text), avoid repeating the same reference multiple times in a paragraph, values in tables might be better separated by brackets e.g. .137 (17.0), some mistaken capitalization, placement of footnote 1 is strange, missing spacing around some references, no space between tables and table legends, paragraph discussing figures is too far away from the figures themselves. I implore the authors to address these points in their next revision. I also agree that providing the most general possible form of Theorem 2, even if in supplementary material, would be ideal.

With this in mind, I give a tentative recommendation for acceptance.

**Additional Comments On Reviewer Discussion:**

This paper experience significant discussion, mostly in response to concerns from Reviewer SGBZ, who raised a perceived error in the proof. The authors corrected the reviewer in the subsequent rebuttal, with the reviewer acknowledging the proof is correct. Some followup discussion considered the assumptions of Theorem 2, making the result more general, but ensuring the Polish assumption remained. Reviewer GLJM provided the most negative review, citing concerns with the complexity reduction claim, which the authors addressed in their rebuttal. Reviewer 4f46 also had negative impressions of the work, citing strong assumptions on Theorem 2. The authors stated that these assumptions could be relaxed significantly. The reviewer appreciated this change, but did not update their score. Reviewers DrUo and SGBZ finished the discussion period with confident positive scores. The authors made several alterations in accordance with Reviewer DrUo's feedback, but initially disagreed with the inclusion of confidence intervals. Reviewer DrUo persisted in ensuring that the authors would include confidence intervals, as per conference guidelines.

---

### Decision · Program_Chairs · 2025-01-22

Accept (Poster)